# Small-molecule G-quadruplex stabilizers reveal a novel pathway of autophagy regulation in neurons

Jose F Moruno-Manchon[1], Pauline Lejault[2], Yaoxuan Wang[1], Brenna McCauley[3], Pedram Honarpisheh[4,5], Diego A Morales Scheihing[4], Shivani Singh[6], Weiwei Dang[3], Nayun Kim[6], Akihiko Urayama[4,5], Liang Zhu[7,8], David Monchaud[2], Louise D McCullough[4,5], Andrey S Tsvetkov[1,5,9]*

[1]Department of Neurobiology and Anatomy, The University of Texas McGovern Medical School at Houston, Houston, United States; [2]Institut de Chimie Moléculaire (ICMUB), UBFC Dijon, CNRS UMR6302, Dijon, France; [3]Huffington Center on Aging, Baylor College of Medicine, Houston, United States; [4]Department of Neurology, The University of Texas McGovern Medical School at Houston, Houston, United States; [5]The University of Texas Graduate School of Biomedical Sciences, Houston, United States; [6]Department of Microbiology and Molecular Genetics, The University of Texas McGovern Medical School at Houston, Houston, United States; [7]Biostatistics and Epidemiology Research Design Core Center for Clinical and Translational Sciences, The University of Texas McGovern Medical School at Houston, Houston, United States; [8]Department of Internal Medicine, The University of Texas McGovern Medical School at Houston, Houston, United States; [9]UTHealth Consortium on Aging, The University of Texas McGovern Medical School at Houston, Houston, United States

*For correspondence:
Andrey.S.Tsvetkov@uth.tmc.edu

Competing interests: The authors declare that no competing interests exist.

**Abstract** Guanine-rich DNA sequences can fold into four-stranded G-quadruplex (G4-DNA) structures. G4-DNA regulates replication and transcription, at least in cancer cells. Here, we demonstrate that, in neurons, pharmacologically stabilizing G4-DNA with G4 ligands strongly downregulates the *Atg7* gene. *Atg7* is a critical gene for the initiation of autophagy that exhibits decreased transcription with aging. Using an in vitro assay, we show that a putative G-quadruplex-forming sequence (PQFS) in the first intron of the *Atg7* gene folds into a G4. An antibody specific to G4-DNA and the G4-DNA-binding protein PC4 bind to the *Atg7* PQFS. Mice treated with a G4 stabilizer develop memory deficits. Brain samples from aged mice contain G4-DNA structures that are absent in brain samples from young mice. Overexpressing the G4-DNA helicase Pif1 in neurons exposed to the G4 stabilizer improves phenotypes associated with G4-DNA stabilization. Our findings indicate that G4-DNA is a novel pathway for regulating autophagy in neurons.

## Introduction

G-quadruplex-DNA (G4-DNA) is a higher-order nucleic acid structure formed by guanine (G)-rich sequences. Co-planar associations of four guanines into G-quartets self-stack to form highly thermo-dynamically stable G4-DNA complexes, which are further stabilized by potassium cations. These structures are important in DNA replication, telomere maintenance, and regulation of transcription, at least in cancer cells (*Rhodes and Lipps, 2015*; *Maizels and Gray, 2013*). Putative G4-DNA form-ing sequences (PQFSes) are ubiquitous in the human genome: more than 300,000 PQFSes have been identified *in silico* and more than 700,000 G4-DNA sequences by G4-seq (*Chambers et al.,*

2015). These sequences are frequent in oncogenes and regulatory and homeostatic genes (*Eddy and Maizels, 2006*; *Huppert and Balasubramanian, 2007*). Intriguingly, the number of the G4-DNA structures varies between cancerous cell lines, indicating that 'active' G4-DNA structures and G4-DNA landscapes might be cell-type dependent (*Hänsel-Hertsch et al., 2016*).

The importance of G4-DNA in cellular homeostasis has been further supported by the discovery of G4-DNA binding proteins. Various proteins, including G4-DNA unwinding helicases (*Sauer and Paeschke, 2017*) (*e.g.,* Pif1 *Paeschke et al., 2013*) and several transcription factors (*Lopez et al., 2017*; *Gao et al., 2015*; *Kumar et al., 2011*), bind to the G4-DNA structures and, therefore, may regulate transcription of specific genes. G4-DNA downregulates gene expression by preventing transcription factor binding to the gene promoter or stalling RNA polymerase. Stabilized G4-DNA must be unfolded for transcription to occur. In contrast, the G4-DNA structures may enhance the expression of certain genes by facilitating transcription factor binding to these genes or their promoters (*Bochman et al., 2012*; *Kumar et al., 2008*; *Smestad and Maher, 2015*) or by keeping the gene 'open' and, thus, enabling re-initiation of transcription (*Bochman et al., 2012*; *Smestad and Maher, 2015*; *Du et al., 2008*; *David et al., 2016*).

Recently, we demonstrated that PQFSes are located in the promoter region of the *Brca1* gene and in the *Brca1* gene itself and that pharmacologically stabilizing G4-DNA downregulates *Brca1* gene and promotes DNA damage in neurons (*Moruno-Manchon et al., 2017*). However, whether G4-DNA regulates gene expression of other genes in highly transcriptionally active neurons is not known. Additionally, G4-DNA was recently implicated in neurodegenerative disorders, such as frontotemporal dementia and amyotrophic lateral sclerosis (*Haeusler et al., 2016*). In aged cells, intriguingly, guanines within DNA are often oxidized, and oxidation stabilizes G-quadruplexes (*Gros et al., 2007*), therefore making these non-canonical structures an attractive research target in neurodegeneration and brain aging research.

Macroautophagy (referred to as autophagy hereafter) is a fundamental cellular process by which cells sequester and degrade proteins, damaged or unwanted organelles, and parasites (*Galluzzi et al., 2017*). Thus, autophagy is critical for cell survival and maintenance, development, inflammation and immune responses, DNA repair, proteostasis, organelle quality control, and prevention of cellular senescence and aging (*Galluzzi et al., 2017*). Mice with enhanced basal autophagy exhibit increased healthspan and lifespan (*Fernández et al., 2018*), but those with defective autophagy develop neurodegenerative disease–like symptoms, indicating that autophagy plays a vital role in neural maintenance and survival (*Komatsu et al., 2006*). To sequester cytoplasmic content, autophagy involves the use of autophagosomes, double-membrane vesicles, which subsequently fuse to lysosomes for degradation (*Galluzzi et al., 2017*). Autophagy is orchestrated by the autophagy-related (ATG) evolutionarily conserved genes that nucleate the autophagosomal precursor phagophore and elongate the autophagosome, engulf cytoplasmic cargo, and fuse the autophagosome with the lysosome (*Galluzzi et al., 2017*). Autophagy is regulated by transcription and translation, as well as by protein post-translational modifications and autophagic proteins' half-lives (*He and Klionsky, 2009*; *Lubas et al., 2018*). A decrease in autophagic activity with aging leads to the accumulation of damaged and senescent cellular components in all cell types of aging organisms (*Cuervo, 2008*). The expression of many critical autophagic genes, such as *Atg5* and *Atg7*, decreases with aging (*Lipinski et al., 2010*; *Lu et al., 2004*), which can also be epigenetically regulated, at least in part (*Lapierre et al., 2015*; *Füllgrabe et al., 2014*). Intriguingly, G4 ligands *stimulate* autophagy in cancer cells (*Beauvarlet et al., 2019*; *Orlotti et al., 2012*; *Zhou et al., 2009*). Whether G4-DNA structures can regulate autophagy in neurons or are altered with aging is not known.

ATG7, an E1-like enzyme, critical for the initiation of autophagy, couples LC3-I to the E2-like enzyme ATG3 leading to the E3-like complex of ATG16L1/ATG5-ATG12 to conjugate LC3-I to phosphatidylethanolamine in phagophore membranes (*Galluzzi et al., 2017*). Mice deficient in genes involved in the ATG conjugation system, including *Atg7*, die within 1 day after birth because autophagy is strongly upregulated immediately after birth as an adaptation mechanism (*Kuma et al., 2017*). Models of neurodegeneration, such as alpha-synucleinopathy and brain samples from patients with Lewy Body disease, show that ATG7 is downregulated, reflecting reduced and defective autophagy, and endogenously raising ATG7 by a lentiviral delivery decreases the levels of alpha-synuclein and mitigates neurodegeneration (*Crews et al., 2010*). *Atg7*-deficient neurons in the midbrain of conditional *Atg7* knock-out mice degenerate and are accompanied by the formation

of ubiquitinated inclusion bodies (*Friedman et al., 2012*). Importantly, the expression of *Atg7* goes down in the human brain during normal aging (*Lipinski et al., 2010*). It is not clear what mechanisms regulate *Atg7* expression.

In this study, we investigated whether G4-DNA regulates neuronal autophagy. We discovered that stabilizing G4-DNA with two distinct G4-DNA-binding ligands, pyridostatin (PDS) and BRACO19, downregulates the ATG7 protein, lowers *Atg7* mRNA, and inhibits autophagy in cultured primary neurons. We also found that, in an in vitro gel-shift assay, an antibody specific to the G4-DNA binds to a synthetic oligonucleotide, which corresponds to a G4-forming sequence in the first intron of the *Atg7* gene. The G4-DNA-binding protein PC4 also binds to this oligonucleotide from the *Atg7* gene. We discovered that mice treated with PDS exhibit memory deficits and accumulate lipofuscin, a hallmark of aged brains. Brain samples from aged mice contained G4-DNA structures that are not present in brain samples from young mice. In cultured primary neurons exposed to PDS, overexpressing the G4-DNA helicase Pif1, a G4-DNA helicase that unwinds the G-quadruplex structures even in the presence of G4-DNA-binding drugs (*Zhou et al., 2014*), improves autophagic phenotypes induced by PDS treatment. Our findings suggest that the G4-DNA structures might be an important pathway during brain aging and neurodegeneration.

## Results

### Autophagic genes contain PQFSes
We hypothesized that many autophagy genes can be regulated by G4-DNA. First, we investigated whether the ATG genes contain putative G4-DNA motifs. We used the QGRS mapper (http://bioinformatics.ramapo.edu/QGRS/index.php) to identify the PQFSes in these genes (mouse, rat and human). Analyses revealed that all these genes contain PQFSes, suggesting that G4-DNA may be involved in the regulation of their expression (*Figure 1*).

### PDS and BRACO19 downregulate *Atg7* in neurons
ATG7 is important for autophagosome biogenesis (*Galluzzi et al., 2017*). The rat *Atg7* gene contains 27 putative sequences that can arrange into G4-DNA. There are no PQFSes in the *Atg7* gene promoter (upstream, 5 kb) (*Figure 2a*). We first determined if PDS alters *Atg7*'s mRNA levels in primary cultured neurons (*Figure 2b*). After PDS treatment, mRNA was extracted and analyzed by qRT-PCR. We found that the levels of *Atg7*'s mRNA were sevenfold lower in neurons exposed to the G4 ligand than in neurons treated with a vehicle. *Tbp* (TATA-binding protein) mRNA was used as loading control as neither *Tbp* nor its promoter contains a PQFS (*Moruno-Manchon et al., 2017*). We next tested if the levels of the ATG7 protein are changed in neuronal cells treated with PDS. Cultured neurons were treated with PDS, and cellular extracts were analyzed by western blotting. The levels of the ATG7 protein in PDS-treated neurons were half those of control neurons (*Figure 2c,d*). We confirmed these findings with another well-established G4 ligand, BRACO-19 (*Haider et al., 2011*). Similarly, BRACO-19 reduced the levels of *Atg7* mRNA and ATG7 by twofold in neurons (*Figure 2e,f,g*).

### A PQFS in the *Atg7* gene folds into a G4 motif in vitro and in vivo
G4-DNA sequences have been extensively studied in vitro. We examined whether the sequence discovered in the *Atg7* gene folds into a G4 structure in vitro. We first identified a 32-nt sequence with the highest QGRS score (G-score = 67; see Materials and methods) in the *Atg7* gene (*Figure 3a*). Whether this sequence, named Atg7-32 (d[$^{5'}$G$_3$GCTGG$_3$TC$_3$T$_2$GG$_3$A$_2$CTGTAT$_2$G$_3^{3'}$]), is able to fold into a G4 structure (*Figure 3b*) was investigated by circular dichroism (CD) and thermal difference spectra (TDS) (*Mergny et al., 2005*). CD and TDS signatures clearly indicated that Atg7-32 indeed folds into a mixture of different topological quadruplex structures (*Figure 3c,d*). These signatures were expected in view of the nature of the intervening sequences between the guanine-runs (from 2-nt to 9-nt loops), which might also form duplex stems. The variety of the Atg7-32 G4-DNA structures can be reduced by dehydrating conditions with PEG200 and CH$_3$CN (*Buscaglia et al., 2013*), leading to the typical CD (negative at 242 nm and positive at 264 nm) and TDS (positive at 273 nm and negative at 296 nm) signatures of a G4-DNA structure (*Figure 3c,d*). Similar experiments were performed with a modified Atg7-32 sequence (named mutAtg7-32) that cannot fold into a G4-DNA

## Autophagy-related genes

| Gene | | Mouse | | Rat | | Human | |
|---|---|---|---|---|---|---|---|
| | | Number of G4 | | Number of G4 | | Number of G4 | |
| Yeast | Mammals | Promoter* | Gene | Promoter* | Gene | Promoter* | Gene |
| Atg1 | ULK1/2 | 0/0 | 9/4 | 0/0 | 3/5 | 6/0 | 41/4 |
| Atg2 | ATG2A/B | 0/1 | 15/7 | 1/0 | 7/11 | 1/1 | 27/1 |
| Atg3 | ATG3 | 1 | 2 | 0 | 0 | 0 | 1 |
| Atg4 | ATG4A/B/C/D | 0/2/0/0 | 28/3/3/1 | **/2/0/0 | **/2/3/2 | 1/2/0/3 | 7/9/1/5 |
| Atg5 | ATG5 | 3 | 10 | 0 | 9 | 0 | 16 |
| Atg6 | BECN1 | 1 | 2 | 0 | 2 | 2 | 4 |
| Atg7 | ATG7 | 0 | 19 | 0 | 27 | 2 | 34 |
| Atg8 | MAP1LC3B | 0 | 5 | 2 | 4 | 0 | 5 |
| Atg9 | ATG9A/B | 0/0 | 4/1 | 0 | 5/5 | 0/2 | 3/15 |
| Atg10 | ATG10 | 0 | 28 | 2 | 31 | 1 | 30 |
| Atg12 | ATG12 | 2 | 0 | 3 | 0 | 4 | 1 |
| Atg13 | ATG13 | 2 | 7 | 0 | 2 | 0 | 2 |
| Atg14 | ATG14 | 0 | 4 | 2 | 2 | 0 | 6 |
| Atg16 | Atg16L1/L2 | 2/1 | 3/3 | 0/0 | 2/3 | 0/1 | 6/10 |
| Atg18 | WIPI1/2 | 1/2 | 18/11 | 3/1 | 7/5 | 5/2 | 8/4 |

**Figure 1.** PQFS in the gene and the promoter sequence of autophagy genes. The number of PQFS in the listed genes and their promoter were analyzed by using the QGRS mapper (http://bioinformatics.ramapo.edu/QGRS/index.php). * 5000 nucleotides upstream the gene were considered as the promoter sequence; ** Data not available.

The online version of this article includes the following source data for figure 1:

**Source data 1.** PQFS in the gene and the promoter sequence of autophagy genes.

structures because of seven G-to-C replacements (underlined) within the four G-runs of the Atg7-32 sequence (d[$^{5'}$GCGCCTGCGCTC$_3$T$_2$GCGCA$_2$CTGTAT$_2$GCG$^{3'}$]). We found that mutAtg7-32 display signatures typical of a GC-rich duplex (CD signals at 255 (negative) and 285 nm (positive); TDS signals at 240 and 276 nm), thus confirming the G4 topological unicity of Atg7-32 (*Figure 3c,d*). We further investigated the higher-order structure of both Atg7-32 and mutAtg7-32 by nuclear magnetic resonance (NMR). Both displayed [1]H-NMR signals in the 12–14 ppm region, which corresponds to duplex stems (providing a rationale for the complicated CD/TDS signature of the former), but only Atg7-32 had [1]H-NMR signals in the 10–12 ppm region, characteristic of a G4-DNA structure (poorly defined here, demonstrating a mixture of G4 topologies) (*Figure 3e*). These signals indicate that Atg7-32 may fold into a variety of G4-DNA topologies, including both 3- and 4-G-quartet G4s with both short (2-nt) and long (9-nt) hairpin-forming loops (*Figure 3b*), which were also detected earlier in non-neuronal cells (*Chambers et al., 2015*; *Puig Lombardi et al., 2019*) or computationally predicted (*Bedrat et al., 2016*; *Puig Lombardi and Londoño-Vallejo, 2020*). An equilibrium among all these various topologies is illustrated by the complex signatures generated with CD, TDS and NMR.

We next investigated whether and how PDS and BRACO-19 (*Figure 3f*) interact with Atg7-32 and mutAtg7-32 in vitro. We found that both ligands strongly stabilize the Atg7-32 G4 structure against thermal denaturation (*via* a FRET-melting assay, using doubly labeled Atg7-32 and mutAtg7-32 sequences, see Methods), delaying melting by 22°C and 14°C for PDS and BRACO-19, respectively, while interacting only moderately with the mutAtg7-32 hairpin structure, delaying its melting by 3°C and 4°C only for PDS and BRACO-19, respectively (*Figure 3g,h*). Overall, these data confirm that Atg7-32 folds into a G4-DNA structure in vitro that can be stabilized by G4-ligands.

A longer version of the Atg7-32 motif, with 6-nt extensions on both its 5′- and 3′-ends, named ATG2700 (d[$^{5'}$AT$_2$CT$_2$G**$_3$**GCTGG**$_3$**TC$_3$T$_2$GG**$_3$**A$_2$CTGTAT$_2$G**$_3$**TGA$_2$C$_2$$^{3'}$]), was used to assess whether it

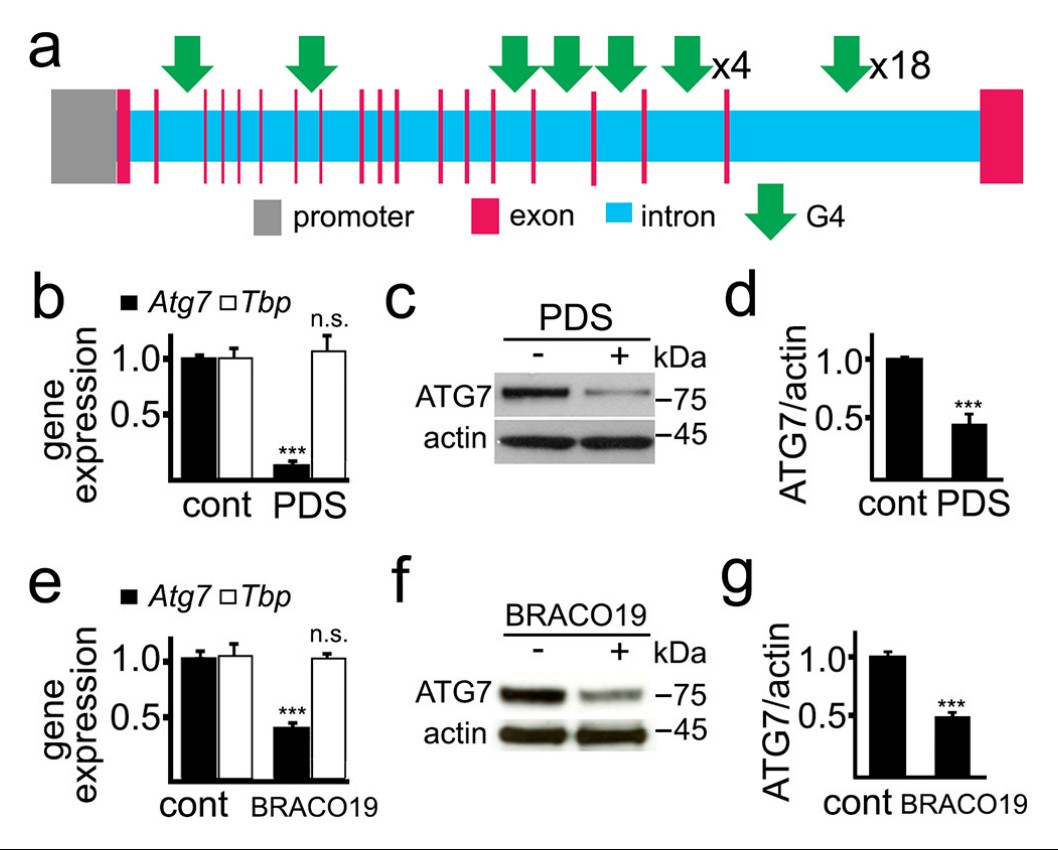

**Figure 2.** PDS downregulates ATG7 levels in primary neurons. (**a**) A scheme of the rat *Atg7* gene and its promoter showing putative G4-DNA locations. (**b–d**) Cultured primary neurons were treated with a vehicle (control, cont) or with PDS (2 µM) overnight. Neurons were collected and processed to measure mRNA (**b**) and levels of ATG7 (**c,d**). (**b**) Expression levels of *Atg7* and *Tbp* (housekeeping protein as control) were determined by qRT-PCR. ***p(*Atg7*)=0.0001 (t-test). n.s., non-significant, p(*Tbp*)=0.426. Results were pooled from three independent experiments. (**c**) The protein levels of ATG7 were determined by western blotting. Actin was used as a loading control. (**d**) Quantification of ATG7 protein levels normalized to actin from (**c**). ***p=0.0001 (t-test). Results were pooled from four independent experiments. (**e–g**) Cultured primary neurons were treated with a vehicle (control, cont) or with BRACO19 (2 µM) overnight. Neurons were collected and processed to measure mRNA (**e**) and levels of ATG7 (**f,g**). (**e**) The expression of *Atg7* and *Tbp* was determined by qRT-PCR. ***p(*Atg7*)=0.0001 (t-test). n.s., non-significant, p(*Tbp*)=0.662. Results were pooled from three independent experiments. (**f**) Levels of ATG7 were determined by western blotting. Actin was used as a loading control. (**g**) Quantification of ATG7 protein was normalized to actin from (**f**). ***p=0.0001 (t-test). Results were pooled from four independent experiments.

can be recognized by the G4-specific antibody HF2 (*Figure 4a*). We synthesized both Cy5-labeled ATG2700 and SS-DNA, a control that cannot fold into a G4 structure. The HF2 antibody was incubated with ATG2700 and SS-DNA in buffers with either K⁺, which favors G4, or Li⁺, which prevents G4 formation (*Figure 4a,b*). We found that HF2 interacts with ATG2700 only in K⁺-rich conditions (*Figure 4a*), without binding to the control SS-DNA. Our data thus indicate that the PQFS identified in the *Atg7* gene indeed adopts a G4 structure in vitro. These findings were further confirmed with a well-established G4-binding protein, PC4. Yeast PC4 (Sub1) and human PC4 (hPC4) were overexpressed in yeast and lysates were incubated with ATG2700 and SS-DNA, immobilized on beads. Yeast and human PC4 only interact with ATG2700 (*Figure 4c,d*), further demonstrating the G4 nature of the *Atg7*'s G4.

Finally, to confirm that the G4s can be detected in vivo, cultured primary neurons were treated with a vehicle or PDS or BRACO19 and then stained with N-TASQ, a G4-DNA-selective fluorophore that has been used to gauge the changes in a G4 landscape in cancer cells treated with G4 ligands (*Laguerre et al., 2015*; *Laguerre et al., 2016*; *Yang et al., 2017*). We discovered that PDS- and

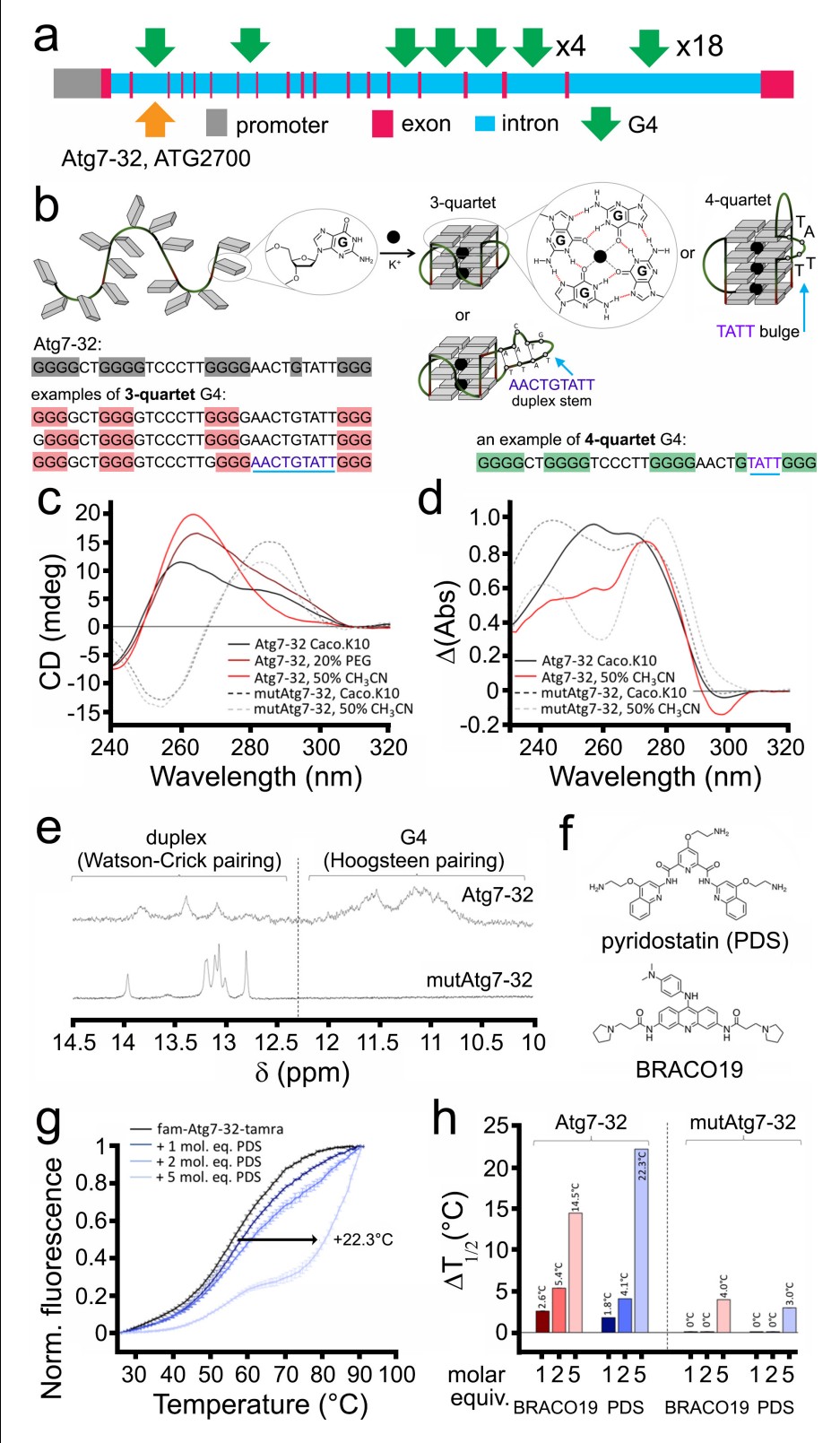

**Figure 3.** A PQFS in the *Atg7* gene folds into a G4 structure in vitro. (a) Scheme of the rat *Atg7* gene and its promoter showing the sequence of the Atg7-32 and ATG2700 oligonucleotides that corresponds to a putative G4-forming sequence. (b) Scheme of the G-rich sequence under its unfolded (left) and folded structures (G4-DNA, right); guanines are shown as gray squares, with detailed chemical structures of guanine (left) and G-quartets

*Figure 3 continued on next page*

*Figure 3 continued*

(right). Atg7-32 may fold into multiple conformations that include both 3- and 4-G-quartet G4s with both short (2-nt) and long (9-nt) hairpin-forming loops. An equilibrium between all these various topologies is illustrated by the complex signatures generated with CD, TDS and NMR (see **c–e**). (**c**) Circular dichroism (CD) generated from 3 µM Atg7-32 (plain lines) and mutAtg7-32 (dotted lines) in 10 mM lithium cacodylate buffer plus 10 mM KCl and 90 mM LiCl (Caco.K10) in absence (black lines) or presence dehydrating agent (PEG200, 20% v/v, brown line, or acetonitrile, 50% v/v, red plain line and gray dotted line for Atg7-32 and mutAtg7-32, respectively). (**d**) Thermal difference spectra (TDS) generated from 3 µM Atg7-32 (plain lines) and mutAtg7-32 (dotted lines) in Caco.K10 in absence (black lines) or presence of acetonitrile (50% v/v, red plain line and gray dotted line for Atg7-32 and mutAtg7-32, respectively). (**e**) Nuclear magnetic resonance (NMR) of 200 µM Atg7-32 (upper panel) and mutAtg7-32 (lower panel) in Caco.K10. (**f**) Chemical structures of PDS and BRACO19. (**g–h**) FRET-melting curves (**g**) and results (**h**) for experiments performed with 0.2 µM fam-Atg7-32-tamra (**g,h**) and fam-mutAtg7-32-tamra (**h**) in absence (black line) or presence of increasing concentrations of PDS (0.2–1.0 µM, blue lines), (**g,h**) and BRACO19 (**h**) in CacoK.10.

BRCAO19-treated neurons exhibit higher levels of N-TASQ fluorescence than control cells (*Figure 4e,f*, and *Figure 4—figure supplement 1*). These data indicate that G4 ligands modulate a G4 landscape in cultured primary neurons, suggesting a mechanism of how G4 stabilization downregulates *Atg7*.

## PDS inhibits neuronal autophagy

To confirm that autophagy is downregulated by PDS, we measured autophagic flux in live neurons. We used an optical pulse-chase labeling method based on the photoswitchable protein Dendra2 and longitudinal imaging (*Tsvetkov et al., 2013a*; *Barmada et al., 2014*; *Tsvetkov et al., 2013b*). Brief irradiation with short wavelength visible light ('photoswitch') irreversibly changes the conformation of 'green' Dendra2 and its fluorescence to the 'red' form (*Figure 5a*). The Dendra2-based optical pulse-chase labeling has been applied to study autophagic flux (*Barmada et al., 2014*; *Moruno Manchon et al., 2015*), protein degradation (*e.g.,* wild-type and mutant huntingtin) (*Tsvetkov et al., 2013b*), the dynamics and turnover of synaptic proteins (*Wang et al., 2009*), and mitochondrial dynamics (*Pham et al., 2012*). Cultured cortical neurons were transfected with Dendra2-LC3 (LC3 is a marker of autophagy *Klionsky et al., 2016*; *Mizushima et al., 2010*), photoswitched, treated with PDS or vehicle, and followed with an automated microscope for several days. The red fluorescence intensities from individual cells were measured at different time points. Decay of the red fluorescence were plotted against time, transformed into log values; the half-lives from individual neurons were analyzed and normalized. Expectedly, the half-life of Dendra2-LC3 (*e.g.,* the decay of photoswitched 'red' Dendra2 signal) was prolonged by PDS by 1.7-fold, indicating slowed flux through autophagy (*Figure 5b*). Beclin1 (*Zhong et al., 2009*), a constitutive protein within the pre-autophagosomal complex used as a positive control, reduced the Dendra2-LC3 half-life by two-fold, indicating that the flux through autophagy was increased, as expected (*Figure 5b*).

We previously discovered a series of small molecules that induce autophagy in primary neurons (*Tsvetkov et al., 2010*). Among them, the benzoxazine derivative 10-NCP promotes neuronal autophagy and protects neurons from misfolded proteins (*Moruno Manchon et al., 2015*; *Tsvetkov et al., 2010*; *Moruno-Manchon et al., 2018*). This compound enhances the formation of autophagosomes and stimulates the lipidation of LC3-I to LC3-II, reflecting enhanced autophagy (*Tsvetkov et al., 2010*; *Moruno-Manchon et al., 2018*). We, therefore, wondered if G4 ligands could reduce and/or prevent 10-NCP-induced lipidation of LC3-I. Primary cortical neurons were treated with PDS, with or without 10-NCP (*Figure 5c,d*). We discovered that PDS completely prevented 10-NCP-mediated formation of LC3-II. In addition, we used BRACO19 alone or in combination with 10-NCP to confirm if lipidation of LC3-II is also inhibited in neurons by an alternative G4 ligand (*Figure 5—figure supplement 1a,b*). BRACO19 reduced the LC3-II levels by 0.8-fold, leading to the conclusion that the initial stages of autophagy are inhibited by G4 ligand treatment, which likely arise from downregulated levels of ATG7, at least in part.

10-NCP, as an autophagy enhancer, regulates the degradation of mutant huntingtin (mHtt) (*Tsvetkov et al., 2010*) in neurons, the protein that causes Huntington's disease. To confirm that PDS modulates autophagic substrates, we transfected neurons with the exon-1 fragment of polyQ-

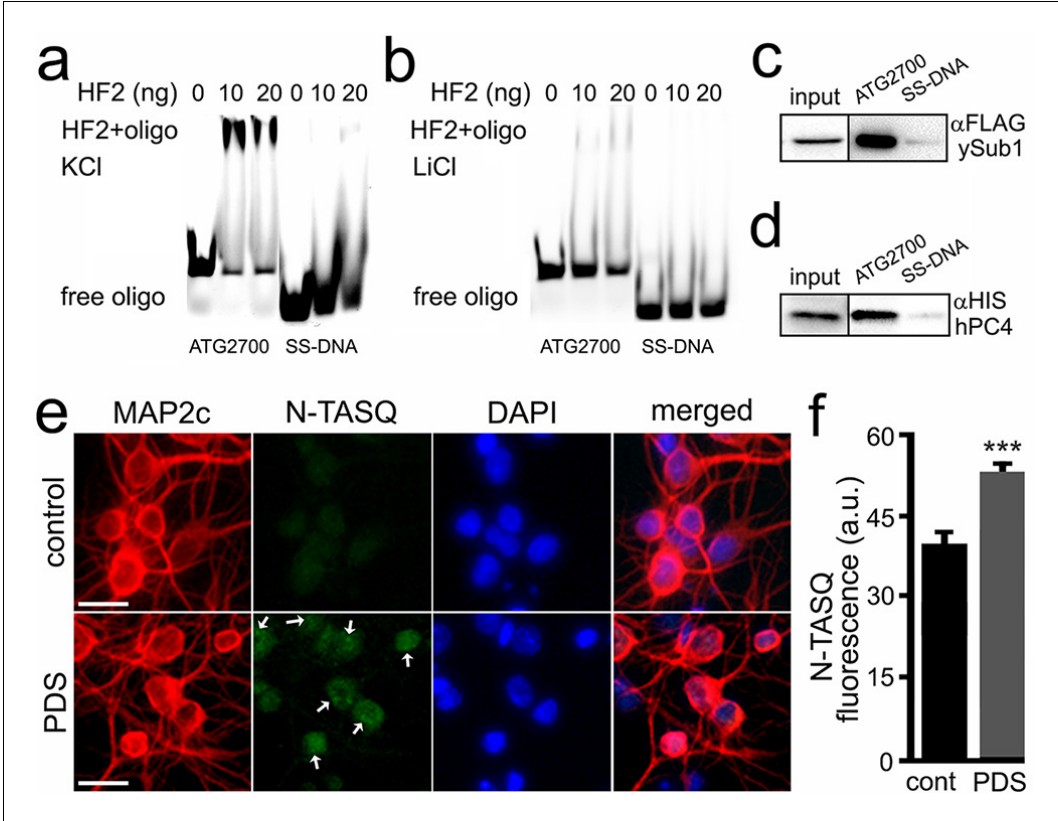

**Figure 4.** The HF2 antibody and PC4 bind to the ATG2700 oligonucleotide in vitro, and the N-TASQ probe detects G4-DNA in vivo. (a–b) Cy5-conjugated ATG2700 and SS-DNA (a negative control) oligonucleotides were heat-denatured and then slow-cooled in the presence of $K^+$ (KCl) or $Li^+$ (LiCl) to allow the formation of a secondary structure. 1.5 pmoles of each oligonucleotides (oligo) and 0 (a buffer alone), 10 or 20 ng of the HF2 antibody were incubated in a buffer, which contained 100 mM KCl (a) or 100 mM LiCl (b). Note in (a) that the bands at the top of the gel correspond to the ATG2700 oligonucleotide bound to the HF2 antibody in samples incubated with a buffer containing KCl. However, note in (b) that the gel lacks of bands at the top. (c–d) Yeast were transformed with a DNA construct that express yeast Sub1-FLAG (c) or with a DNA construct that express human PC4-HIS (d). Yeast were collected and lysed, and extracts were incubated with ATG2700 or SS-DNA (negative control) oligonucleotides. Immunoprecipitates were immobilized with agarose beads, and protein complexes were then run in a gel and analyzed by western blotting with antibodies against FLAG (c) or antibodies against HIS (d). (e) Cultured primary neurons were treated with a vehicle (control, cont) or with PDS (2 µM) overnight. Cells were fixed and stained with N-TASQ (50 µM), with antibodies against MAP2c, and with the nuclear dye Hoechst (DAPI). White arrows depict N-TASQ-positive puncta. Scale bar, 10 µm (f) N-TASQ fluorescence intensities were analyzed from (e). ***p(cont vs PDS)=0.0001 (t-test). For each experiment, 200 neurons were analyzed, and results were pooled from three independent experiments.

The online version of this article includes the following figure supplement(s) for figure 4:

**Figure supplement 1.** BRACO19 changes a G4 landscape in cultured primary neurons.

expanded mHtt (mHtt[ex1]) tagged with Dendra2 and treated them with PDS or vehicle. Importantly, we confirmed that neither the plasmid promoter (pGW1 *Arrasate et al., 2004*) nor the mHtt[ex1] contain putative G4s with QGRS mapper analyses. We found that the half-life of mHtt[ex1]-Dendra2 was increased in neurons exposed to PDS by 1.4-fold (*Figure 5e*). We then used the BACHD mouse model to confirm that PDS affects the degradation of an autophagy substrate, mHtt. BACHD mice express the full-length human mHtt gene and recapitulate multiple features of Huntington disease (*Gray et al., 2008*). We cultured primary cortical neurons from BACHD mouse pups and treated them with a vehicle or PDS. mHtt levels were measured with western blotting. As expected, PDS treatment increased the levels of mHtt by twofold, indicating that degradation of mHtt is inhibited (*Figure 5—figure supplement 2a,b*). Actin was used as a loading control, as we previously found

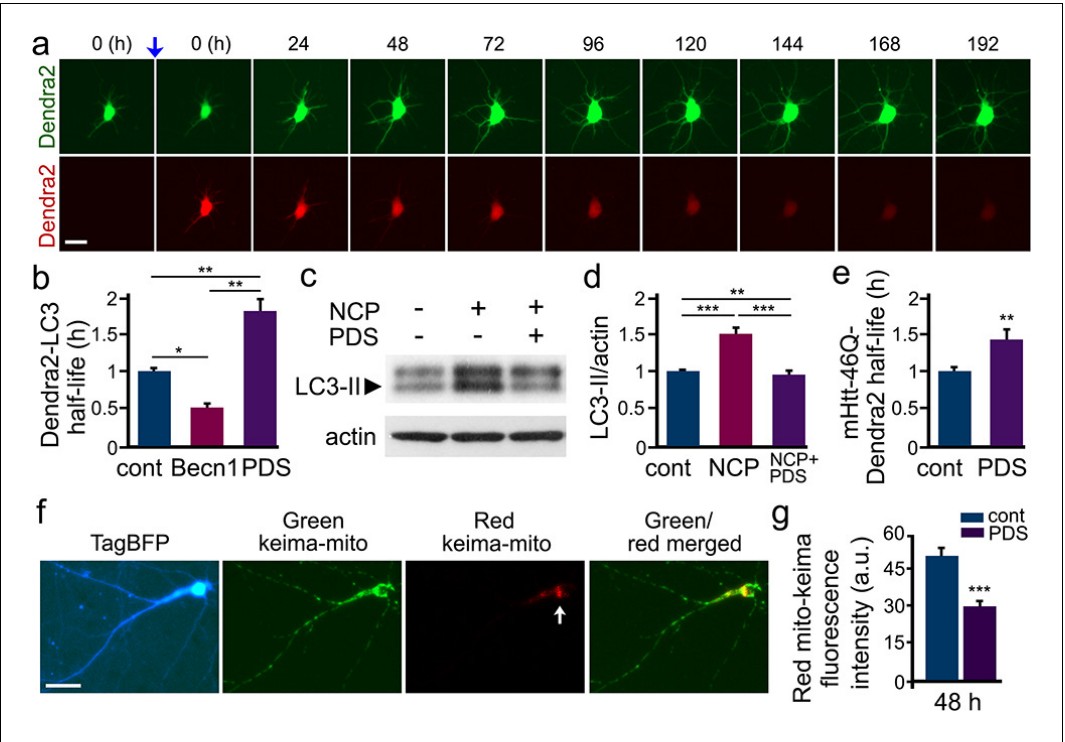

**Figure 5.** PDS inhibits autophagy in cultured primary neurons. (**a**) The photoswitchable protein Dendra2 is commonly used to measure the half-life of a protein of interest. A brief irradiation with short-wavelength visible light induces an irreversible conformational change ('photoswitch', indicated by a blue arrow) in Dendra2. Photoswitched Dendra2 emits red fluorescence that can be tracked overtime with an automated microscope. Scale bar, 10 μm. (**b**) Dendra2 was fused to LC3, an autophagy marker, to measure autophagy flux. Two cohorts of primary neurons were co-transfected with Dendra2-LC3 and an empty plasmid, or with Dendra2-LC3 and untagged beclin1 (Becn1, as a positive control). Neurons co-transfected with Dendra2-LC3 and an empty plasmid were treated with a vehicle (control, cont), or with 0.1 μM PDS overnight. After treatment, neurons were longitudinally imaged, and the decay of the red fluorescence over time was used to calculate the half-life of Dendra2-LC3. The half-life of Dendra2-LC3 is normalized to one with respect to control neurons. *p(cont vs Becn1) =0.02, **p(cont vs PDS)=0.001, **p(Becn1 vs PDS)=0.001 (one-way ANOVA). One hundred neurons per group were analyzed from two independent experiments. (**c**) Cultured primary neurons were treated with a vehicle or with PDS (2 μM), in combination with the autophagy enhancer 10-NCP (NCP, 1 μM) overnight. Neurons were collected, and pellets were lysed and analyzed by western blotting with antibodies against LC3-II and against actin. (**d**) Quantification of LC3-II levels normalized to actin from (**c**). **p(cont vs NCP+PDS)=0.008, ***p(cont vs NCP) =0.0001, ***p(NCP vs NCP+PDS)=0.0002 (one-way ANOVA). Results were pooled from four independent experiments. (**e**) Dendra2 was fused to Htt$^{ex1}$-Q$_{46}$, an autophagy substrate, to measure autophagy flux. Two cohorts of primary neurons were transfected with Dendra2- Htt$^{ex1}$-Q$_{46}$. 24 hr after transfection, neurons were treated with a vehicle (control, cont), or with PDS (0.1 μM), and longitudinally imaged. The decay of the red fluorescence over time was used to calculate the half-life of Dendra2- Htt$^{ex1}$-Q$_{46}$. The half-life of Dendra2- Htt$^{ex1}$-Q$_{46}$ is normalized to one with respect to control neurons. **p(cont vs PDS)=0.0064 (t-test). Fifty neurons per group were analyzed from two independent experiments. (**f**) Fluorescence images of a neuron co-transfected with the DNA constructs TagBFP and mito-Keima. Keima is a fluorescent pH-sensitive protein used as a reporter of subcellular acidic environments. Keima emits green fluorescence in neutral environments, and emits red light in acidic environments, such as lysosomes or autolysosomes. Targeting Keima to mitochondria has been used to study a specific form of autophagy, mitophagy. Note that a white arrow depicts mitochondria in acidic compartment (red channel). (**g**) Two cohorts of primary neurons were transfected with mito-Keima. 24 hr after transfection, neurons were treated with a vehicle (control, cont) or with 0.1 μM PDS, and imaged 48 hr after treatment. Quantification of red fluorescence intensity of mito-Keima indicates that mitophagy is reduced in PDS-treated neurons. ***p(cont vs PDS)=0.0001, (t-test). One hundred neurons per group were analyzed from three independent experiments.

The online version of this article includes the following figure supplement(s) for figure 5:

**Figure supplement 1.** BRACO19 inhibits autophagy in cultured primary neurons.

*Figure 5 continued on next page*

*Figure 5 continued*

**Figure supplement 2.** PDS induces accumulation of mutant huntingtin in cultured primary neurons from Huntington disease mice.

**Figure supplement 3.** ATG7 mitigates autophagy impairment and neurotoxicity associated with PDS treatment in neurons.

that the levels of the actin protein in neurons are not significantly affected by PDS (*Moruno-Manchon et al., 2017*).

Next, we assessed whether G4 ligands regulate a specific form of autophagy, mitophagy, the autophagic degradation of mitochondria that depends on ATG7 (*Vincow et al., 2013*). To measure mitophagy in live neurons, we used an optical method that combines a pH-sensitive protein Keima with automated imaging. Keima is a fluorescent protein that changes both its excitation and emission spectra in response to environmental pH changes, emitting green light at neutral pH and red light at acidic pH. Mitochondrially targeted Keima has been successfully used to study mitophagy (*Katayama et al., 2011*; *Proikas-Cezanne and Codogno, 2011*). Primary cortical neurons were transfected with mito-Keima and BFP, treated with a vehicle, PDS or BRACO19, and the red fluorescence intensity of mito-Keima was analyzed in individual neurons (*Figure 5f,g* and *Figure 5—figure supplement 1c*). Similar to previous studies (*Cai et al., 2012*), we found that basal mitophagy is a relatively slow process, with first mitochondria appearing in the lysosomes ~ 2 days after mito-Keima transfection, and that mitophagy is primarily localized to the neuronal soma (*Figure 5f,g*). In neurons treated with PDS, mitophagy efficiency was reduced by 0.6-fold compared to neurons treated with a vehicle (*Figure 5g*). Interestingly, the BRACO19 treatment reduced mitophagy by 0.8-fold, indicating that PDS and BRACO19 affect neuronal homeostasis differently. Thus, we conclude that, in primary cultured neurons, G4 stabilization downregulates autophagy, including mitophagy.

Finally, we wondered whether ectopic expression of ATG7 mitigates neurotoxicity and autophagy deficits induced by PDS. p62 or sequestosome-1 is a scaffolding protein that acts as an adaptor to identify and deliver cargo to the autophagosome for degradation (*Liu et al., 2017*; *Katsuragi et al., 2015*). p62 is degraded together with the cargo, making p62 a commonly used autophagy marker. Two cohorts of cultured neurons were transfected with p62-GFP and mApple. The third neuronal cohort was transfected with p62-GFP and ATG7-mApple. mApple-expressing neurons were treated with a vehicle (control) or with PDS, neurons transfected with p62-GFP, and ATG7-mApple were

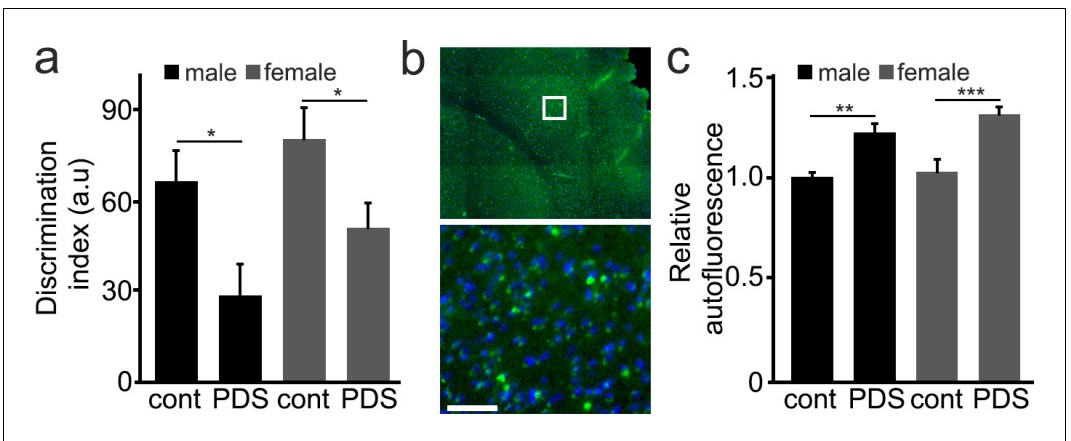

**Figure 6.** Mice treated with PDS develop memory deficits and aged-related symptoms. (**a**) 25-month-old male and female mice were intraperitoneally injected with a solution of a vehicle in PBS (control, cont) or with a solution of PDS in PBS (5 mg/kg, PDS) once a week for 8 weeks. After treatment, mice were tested for short-term memory in the novel object recognition test and discrimination index (DI) was calculated. *p-value(male-cont vs PDS)=0.0265, *p-value(female-cont vs PDS)=0.0382, p-value(male vs female)=0.1029 (two-way ANOVA). Six mice per group were analyzed. (**b**) Mice were sacrificed, and their brains were analyzed for the lipofuscin autofluorescent age pigment. (**c**) Quantification of autofluorescence from (**c**). **p-value(male-cont vs PDS)=0.0043, ***p-value(female-cont vs PDS)=0.0007, p-value(male vs female)=0.2121 (two-way ANOVA). Six mice per group were analyzed.

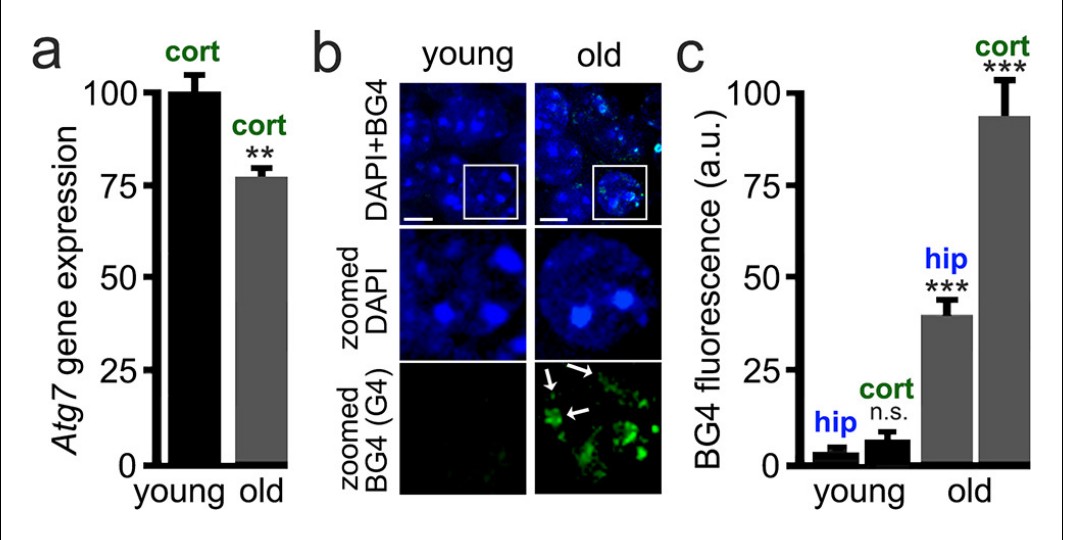

**Figure 7.** Brain samples from aged mice exhibit elevated levels of G4-DNA. 3-month-old (young) and 25-month-old (old) mice were sacrificed, and their brains were processed for RT-qPCR and immunohistochemistry analysis. (a) Cortical brain samples from young and old mice were lysed, and mRNA was extracted. mRNA samples were retro-transcribed and analyzed for expression of the *Atg7* gene. **p(young vs old)=0.0011 (t-test). Six mice per group were analyzed. (b) Brain samples from young and old mice were stained with antibodies against BG4 (green channel) and the Hoechst dye (nuclei marker, blue channel), and imaged with a fluorescent microscope. In the zoomed image, white arrows depict some G-quadruplex-positive structures in the nuclei. (c) Quantification of BG4 fluorescence intensity in the hippocampus (hip) and the cortex (cort) of young and old mice. ***p(hip-young vs old) =0.0001; ***p(cor-young vs old)=0.0001. n.s., non-significant, p(young-hip vs cor)=0.35 (one-way ANOVA). Six mice per group were analyzed.

treated with PDS (PDS+ATG7). We analyzed fluorescence intensity of p62-GFP and discovered that PDS-treated mApple-expressing neurons exhibited a 1.7-fold increase of p62-GFP fluorescence intensity over control neurons. Interestingly, ATG7-overexpressing neurons treated with PDS displayed 0.7-fold reduction of p62-GFP fluorescence intensity, indicating that overexpressing ATG7 mitigates the inhibitory effects of PDS on neuronal autophagy (*Figure 5—figure supplement 3a,b*). Similarly, ATG7 overexpression mitigates PDS-induced neurotoxicity (*Figure 5—figure supplement 3c*). These data further highlight the importance of ATG7 in neuronal autophagy and survival.

## Mice treated with PDS develop memory deficits

We then wondered if PDS would have any effect on the brain in mice. Stabilizers of G4-DNA are being investigated as an anti-cancer therapy. In a prior in vivo study, a G4-binding small molecule (MM41) was used as an anti-cancer therapy with a dosage and schedule that was tolerated (*Ohnmacht et al., 2015*). In our studies, we used a comparable dosage and schedule of PDS. In these experiments, we used old male and female mice (25 months). Mice were randomized and injected weekly with a vehicle or PDS for 8 weeks (4 mg/kg/week), and thereafter, these mice completed the novel object recognition (NOR) test, a standard test for recognition memory that assesses both hippocampal and cortical cognitive function (*Antunes and Biala, 2012*). The discrimination index measures the ability of the tested animal to differentiate a novel object from the familiar object, which was previously presented to the animal. Thus, higher discrimination index indicates if the animal is able to recognize the novel object. Male and female mice treated with PDS exhibited a reduced discrimination index, compared to vehicle-treated mice (*Figure 6a*). These were old mice and even vehicle-treated mice were expected to exhibit significant age-associated neuropathology. We analyzed one of the hallmarks of aging and downregulated autophagy—the levels of lipofuscin—in male and female mice. Lipofuscin is a mixture of accumulated oxidized proteins and lipids found in aged brains (*Brunk and Terman, 2002*). Brains from PDS-treated mice contained more lipofuscin than vehicle-treated mice (*Figure 6b,c*), demonstrating that the treatment with PDS promotes

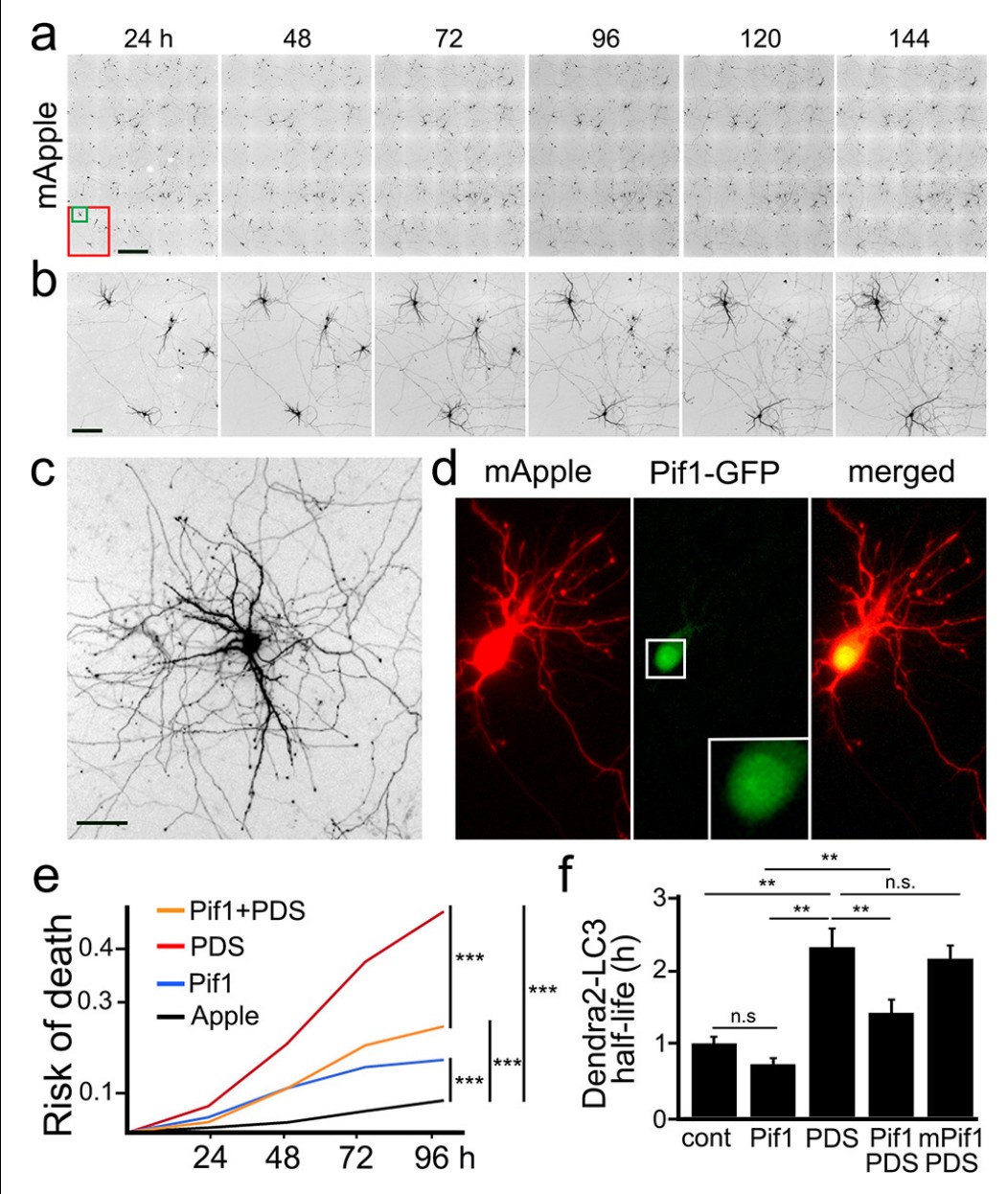

**Figure 8.** The helicase Pif1 restores autophagy in PDS-treated neurons. (**a**) An example of survival analysis. Cultured primary neurons were transfected with the red fluorescent protein mApple as a marker of cell morphology and viability. Neurons were longitudinally imaged each 24 hr for 6 days with an automated microscope. Each image is a montage of 25 individual non-overlapping images. Scale bar is 400 µm. (**b**) Zoomed images from (**a**) at each time point. Images demonstrate the ability to track the same group of neurons over time, and they show the progression of neuron development (note how neurites grow in the top-left neuron), and neurodegeneration (two neurons to the right side of the images gradually lose their neurites until they die). Scale bar is 40 µm. (**c**) Zoomed image that shows the complex neurite arborization of a depicted neuron from (**b**) 144 hr after transfection. Scale bar is 20 µm. (**d**) An example of a neuron co-transfected with mApple (red) and Pif1-GFP (green). Note that Pif1-GFP is mostly nuclear. (**e**) Two cohorts of cultured primary neurons were co-transfected with mApple (as a morphology and survival marker) and GFP (as a control construct), and other two cohorts of neurons were co-transfected with mApple and Pif1-GFP. One cohort of neurons co-transfected with mApple and GFP was treated with a vehicle (Apple), and the second one with 0.5 µM PDS. Another cohort of neurons co-transfected with mApple and Pif1-GFP was treated with a vehicle (Pif1), and the second one with 0.5 µM PDS (Pif1+PDS). Neurons were longitudinally imaged during 4 days, and risks of death were analyzed. ***p=0.0001 (log-rank test). Results were pooled from three independent experiments. (**f**) Two cohorts of cultured primary neurons were co-

*Figure 8 continued on next page*

*Figure 8 continued*

transfected with Dendra2-LC3 and GFP (as a control construct), and other two cohorts of neurons were co-transfected with Dendra2-LC3 and Pif1-GFP. One cohort of neurons co-transfected with Dendra2-LC3 and GFP was treated with a vehicle (control, cont), and the second one with 0.1 µM PDS (PDS). Also, one cohort of neurons co-transfected with mApple and Pif1-GFP was treated with a vehicle (Pif1), and the second one with 0.1 µM PDS (Pif1+PDS). One last cohort of neurons was transfected with mApple and mutant Pif1-GFP and treated with PDS (mPif1+PDS). Neurons were longitudinally imaged, and the half-life of Dendra2-LC3 of each group was analyzed, and normalized to one with respect to the control group. \*\*p(cont vs PDS)=0.001, \*\*p(PDS vs Pif1+PDS)=0.001, p(Pif1 vs Pif1+PDS)=0.001. n.s., non-significant, p(cont vs Pif1)=0.1726, p(cont vs Pif1+PDS)=0.3239, p(PDS vs mPif+PDS)=0.4267 (one-way ANOVA). One hundred neurons per group were analyzed from two independent experiments.

aging phenotypes in the mouse brains. Our data also suggest that anticancer drugs that target G4-DNA may accelerate brain aging and lead to early dementia.

## Brain samples from old mice contain stable G4-DNA

Transcription of many genes is altered in the aged brain (*Lu et al., 2004*), and many of these gene bodies or promoters contain PQFSes. For example, expression of *Atg7* decreases in the human brain during normal aging (*Lipinski et al., 2010*). In yeast, flies, worms, and human immune and cancer cells, histone and chromatin modifications regulate *Atg7* expression (*Settembre et al., 2011*; *Eisenberg et al., 2009*; *Eisenberg et al., 2014*). We first analyzed the mRNA levels of *Atg7* in young and aged mice and observed that *Atg7* mRNA is downregulated by 25% in the aged brains compared to brain samples from young mice (*Figure 7a*). Second, we hypothesized that aged brains contain stable G4-DNA. To test that, we used the BG4 antibody, which recognizes G4 structures in fixed cytological samples (*Biffi et al., 2013*). Young (3 months old) and old (25 months old) mice were sacrificed, and their brains were analyzed by immunohistochemistry with the BG4 antibody. BG4-positive puncta were seen in aged mice and were very rarely seen in young mice (*Figure 7b,c*). These data suggest that the G4 landscape is modulated by aging in vivo, which opens new avenues for aging research.

## Pif1 rescues PDS-induced phenotypes in cultured primary neurons

More than 20 G4 helicases unwind G4-DNA, among which Pif1 is one of the most potent and studied (*Paeschke et al., 2013*). Pif1 unwinds G4 structures even in the presence of G4 ligands (*Zhou et al., 2014*). Thus, we wondered whether Pif1 could rescue PDS-induced phenotypes in cultured primary neurons. Primary neurons were transfected either with GFP and mApple (a marker of viability and morphology) or with Pif1-GFP and mApple (*Figure 8a–d*). Loss of the mApple fluorescence is a sensitive marker of neuronal death (*Tsvetkov et al., 2013a*; *Tsvetkov et al., 2010*; *Arrasate and Finkbeiner, 2005*). Therefore, by analyzing when each neuron lost its fluorescence, we can measure neuronal survival with cumulative hazard statistics (*Figure 8e*). Transfected neurons were tracked longitudinally for several days. Surprisingly, Pif1-GFP was somewhat toxic for primary cortical neurons, in comparison with control GFP-expressing neurons (*Figure 8e*). Nevertheless, Pif1-GFP partially rescued PDS-associated neurotoxicity. Next, we tested if Pif1-GFP rescues autophagic deficits in cultured neurons exposed to PDS. Remarkably, Pif1-GFP reduced the half-life of Dendra2-LC3 by 0.6-fold in neurons treated with PDS (*Figure 8f*). We then used a mutant form of Pif1 without ATPase/helicase activity as a control. Expectedly, mutant Pif1 could not rescue PDS-associated autophagy reduction (*Figure 8f*). These data indicate that Pif1 likely activates coping mechanisms in degenerating neurons, leading to better autophagy.

## Discussion

In this study, we demonstrated that the levels of *Atg7* and, therefore, neuronal autophagy are downregulated by the G4-ligands PDS and BRACO19. We showed that a PQFS identified in the *Atg7* gene can fold into a G4 structure, as demonstrated by spectroscopy (CD, TDS and NMR), which interacts with PDS and BRACO-19, the HF2 antibody, and the G4-binding protein PC4. Mice treated with PDS exhibited memory deficits and accumulation of lipofuscin. Importantly, we discovered that

aged mouse brains contain numerous G4-DNA, while young brains have very few. Our data suggest that an age-associated change in DNA conformation could be a novel epigenetic-like mechanism of gene expression in aging neurons (*Kim, 2019*).

There is a good consensus in the autophagy field that autophagy plays a positive role in slowing aging and increasing longevity (*Hansen et al., 2018*). Autophagy-related genes are critical for longer healthspan and lifespan extension in worms, flies, and mice (*Hansen et al., 2018*). Mice with enhanced basal autophagy have increased healthspan and lifespan (*Fernández et al., 2018*). A decrease in autophagic activity leads to the accumulation of damaged and senescent cellular components in almost all cell types of aging organisms (*Cuervo, 2008*). Transcription factors, such as TFEB and FOXO, regulate the expression of many autophagy genes involved in the healthspan and lifespan (*Lapierre et al., 2015*). We previously demonstrated that transcription factors Nrf2 and TFEB positively regulate neuronal autophagy and promote basal neuronal survival and survival of neurons under stress (*Tsvetkov et al., 2013b*; *Moruno-Manchon et al., 2016*). Epigenetic histone and chromatin modifications also regulate autophagy during aging (*Lapierre et al., 2015*; *Baek and Kim, 2017*). For example, autophagy genes can be epigenetically silenced (*Baek and Kim, 2017*; *Artal-Martinez de Narvajas et al., 2013*). Conversely, pharmacologic inhibition or genetic downregulation of histone methyltransferase G9a leads to the activation of autophagy in cancer cells and in fibroblasts (*Artal-Martinez de Narvajas et al., 2013*). The expression of many critical autophagic genes, such as *Atg5* and *Atg7*, decreases with aging (*Lipinski et al., 2010*; *Lu et al., 2004*). Many of these genes contain PQFS motifs in their introns, exons or promoters. Our findings indicate that G4-DNA may play crucial roles in transcription of autophagic genes in aged neurons.

A link between G4-DNA ligands and autophagy has already been demonstrated in cancer cells. Melanoma cells stop dividing and upregulate autophagy when treated with the G4 ligand Ant1,5 (*Orlotti et al., 2012*). In agreement with this study, a G4 agent, SYUIQ-5, inhibits proliferation, damages G4-DNA enriched telomeres, and upregulates autophagy in CNE2 and HeLa cancer cells (*Zhou et al., 2009*). The G4 ligand 20A causes cell growth arrest and upregulates autophagy in HeLa cells (*Beauvarlet et al., 2019*). In our work, however, we found that, in neurons, G4-ligands trigger opposite effects, downregulating autophagy in post-mitotic neurons, which comes with no surprise as the autophagic pathways in neurons differ from those in other cell types (*Kulkarni et al., 2018*).

G4-DNA-associated regulation of transcription extends well beyond the autophagy genes. We recently demonstrated that PDS and BRACO19 downregulate the *Brca1* gene in cultured primary neurons—the Brca1 gene and gene's promoter contain G4-DNA motifs—leading to DNA damage (*Moruno-Manchon et al., 2017*). Ectopically increasing BRCA1 levels attenuates DNA damage associated with PDS treatment, indicating that *Brca1* downregulation impedes DNA damage repair and DNA double strand breaks accumulate as a result. Age-dependent accumulation of stabilized G4-DNA structures in diverse genes may lead to neuronal senescence and, eventually, to neurodegeneration. In some neurodegenerative diseases and in advanced aging, neurons exhibit various DNA/chromatin abnormalities, including aneuploidy and transposable element dysregulation (*Mosch et al., 2007*; *Fischer et al., 2012*; *Sun et al., 2018*). Our findings suggest that age-dependent changes in DNA conformation and accumulation of G4-DNA could represent a novel mechanism of senescence that includes the autophagic and non-autophagic genes in general. Future studies will determine the exact G4-DNA loci in neurons, how these loci differ between neuronal cell types (*e.g.,* cortical versus granular cerebellar), and how these loci change as neurons develop and age.

G4-DNA structures fold spontaneously within single-stranded DNA (ssDNA) transiently formed during DNA replication, and helicases (including Pif1) unfold them (*Rhodes and Lipps, 2015*). A similar process occurs during transcription on ssDNA in a transcriptional bubble, and Pif1 dismantles these G4-DNA structures as well (*Rhodes and Lipps, 2015*). G4-RNA structures being mostly protein bound form only transiently in living cells (*Fay et al., 2017*; *Yang et al., 2018*). Post-mitotic neurons do not divide, and so, G4 ligands may have a strong effect on co-transcriptionally formed G4-DNA and G4-RNA. In our current and previous studies (*Moruno-Manchon et al., 2017*), we observed no significant accumulation of mRNA (*Atg7* and *Brca1 Moruno-Manchon et al., 2017*) in PDS-treated neurons (*e.g.,* G4-RNA), suggesting that mRNA stability is not considerably affected by G4 ligands. Therefore, in post-mitotic neurons, the primary target of G4-ligands would likely be transcription. Nevertheless, we cannot exclude a possibility that PDS may affect RNA metabolism by stabilizing

G4-RNA. Intriguingly, we found that PDS downregulates *Atg7* stronger than BRACO19, indicating that these two ligands have different affinities towards the G4 structures in living neurons or/and bind to different G4-DNA conformations. As neurons are highly specialized cells, they may have their own, unique G4-DNA pathways, which may be drastically different from G4-DNA mechanisms in non-neuronal cells.

Pif1 is a class of nuclear and mitochondrial 5′−3′ DNA helicases present in all eukaryotes (*Mendoza et al., 2016*). Originally identified in yeast as an important factor for maintaining mitochondrial DNA (*Lahaye et al., 1991*), Pif1's functions now include the regulation of telomere length, replication, and resolving G4-DNA (*Byrd and Raney, 2017*). Among G4-DNA helicases, Pif1 is one of the most potent and can unwind G4-DNA stabilized by G4-DNA-interacting small molecules (*Paeschke et al., 2013*; *Zhou et al., 2014*). We demonstrate that Pif1 rescues phenotypes associated with PDS treatment. Intriguingly, expression of Pif1 itself somewhat upregulates autophagy. Therefore, in addition to histone acetyltransferases facilitating chromatin decondensation and promoting the expression of autophagy-related genes (*Lapierre et al., 2015*; *Baek and Kim, 2017*), Pif1 may also help to sterically allow the transcriptional machinery to transcribe DNA, including autophagic and non-autophagic genes. Nevertheless, we cannot fully exclude a possibility that Pif1 is protective in our experiments with PDS due to unknown functions besides being a G4-DNA helicase. Intriguingly, prior in vitro studies found that Pif1's G4-DNA unwinding activity is diminished by G4 ligands (*e.g.*, PDS), which appears to contradict to our in vivo findings. Nevertheless, the relevance of these data to our study is not straightforward since a G4-DNA forming sequence was used without its complementary sequence in the in vitro studies. Adding the complementary DNA sequence unfolds the G4-DNA/ligand complexes (*Mendoza et al., 2016*). In addition, the in vitro experiments assayed the activity of Pif1 using an excess of G4 ligands (*Mendoza et al., 2016*), and therefore, the data are not easy to extrapolate to our neuronal in vivo model. Also, in our studies with living neurons, Pif1 was overexpressed before PDS was added to the media, and thus, the kinetics of Pif1-G4-DNA-PDS interactions may be overly complex for a direct comparison to the in vitro conditions.

Our findings have important ramifications for aging and neurodegeneration research. We and others previously demonstrated that neuronal autophagy can be targeted therapeutically to mitigate or potentially stop neuronal aging and neurodegeneration. In this study, we demonstrate that there is a novel layer of autophagy regulation – G4-DNA. Our data suggest that G4-DNA and G4-DNA-regulating proteins might be promising therapeutic targets for developing therapies against age-associated neurodegenerative disorders.

# Materials and methods

**Key resources table**

| Reagent type (species) or resource | Designation | Source or reference | Identifiers | Additional information |
|---|---|---|---|---|
| Gene (Rattus norvegicus) | ATG7 | N/A | Gene-NCBI: ID: NC_005103.4 | |
| Strain, strain background (*Mus musculus*) | C57BL/6 | Jackson Laboratoy | 664 | female and male |
| Strain, strain background (Rattus norvegicus) | Long Evans | Charles River | 6 | N/A |
| Cell line (Rattus norvegicus) | primary cortical neurons | | | Neurons isolated from Long-Evans rat embryos (E17–18) cortices |

*Continued on next page*

Continued

| Reagent type (species) or resource | Designation | Source or reference | Identifiers | Additional information |
|---|---|---|---|---|
| Antibody | anti-microubule-associated protein 1 light chain three alpha (LC3; rabbit polyclonal) | MBL | #PD014 | (1:1000), overnight 4°C |
| Antibody | anti-atg7 (clone D12B11; rabbit monoclonal) | Cell Signaling | #8558 | (1:1000), overnight 4°C |
| Antibody | anti-beta actin (clone 8H10D10; mouse monoclonal) | Cell Signaling | #3700 | (1:2000), overnight 4°C |
| Antibody | anti-DYKDDDDK Tag (FLAG; clone D6W5B; rabbit polyclonal) | Cell Signaling | #2368 | (1:500), overnight 4°C |
| Antibody | anti-microtubule-associated protein-2 (MAP-2; clone A-4; mouse monoclonal) | Santa Cruz Biotechnology | #sc-74421 | (1:500), overnight 4°C |
| Antibody | anti-rabbit-HRP | EMD Millipore | #AP307P | (1:2000), overnight 4°C |
| Antibody | anti-mouse-HRP | EMD Millipore | #AP308P | (1:2000), overnight 4°C |
| Antibody | anti-mouse Alexa Fluor-488 | Life Technologies | #A11001 | (1:500), overnight 4°C |
| Antibody | anti-Huntingtin protein (Htt; clone mEM48; mouse monoclonal) | EMD Millipore | #MAB5374 | (1:1000), overnight 4°C |
| Antibody | anti-rabbit Alexa Fluor-546 | Life Technologies | #A11010 | (1:500), overnight 4°C |
| Antibody | anti-HF2 | (*Lopez et al., 2017*) | | (1:100), overnight 4°C, prepared in Nayun Kim's lab |
| Antibody | anti-BG4 | (*Biffi et al., 2013*) | | (1:100), overnight 4°C, prepared in Nayun Kim's lab |
| Recombinant DNA reagent | lipofectamine 2000 | Thermo Fisher Scientific | 12566014 | |
| Recombinant DNA reagent | pCAG-TagBFP | VectorBuilder | pRP[Exp]-CAG > TagBFP | |
| Recombinant DNA reagent | pCAG-EGFP-hPIF1 | VectorBuilder | pRP[Exp]-CAG > EGFP(ns): hPIF1 [ORF026999] | |
| Recombinant DNA reagent | pCAG-EGFP-mutant hPIF1 | VectorBuilder | pRP[Exp]-CAG > EGFP(ns):{hPIF1 [ORF026999]*(E307Q)} | |
| Recombinant DNA reagent | EF1A-mApple-ATG7 | VectorBuilder | pRFP[Exp]-EF1A > mApple(ns):mAtg7 [NM_001253717.1] | |
| Recombinant DNA reagent | pSANG10-3F-BG4 | Addgene | #55756; deposited by Dr. Shankar Balasubramanian, the University of Cambridge | |
| Recombinant DNA reagent | pGW1-Dendra2-LC3 | (*Tsvetkov et al., 2013b*) | | |
| Recombinant DNA reagent | Httex1-Q46-Dendra2 | (*Tsvetkov et al., 2013b*) | | |

*Continued on next page*

*Continued*

| Reagent type (species) or resource | Designation | Source or reference | Identifiers | Additional information |
|---|---|---|---|---|
| Recombinant DNA reagent | pGW1-mito-Keima | other | | It was cloned from the mt-mKeima/pIND(SP1) construct that we kindly received from Dr. Atsushi Miyawaki (RIKEN Brain Science Institute, Japan) |
| Sequence-based reagent | ATG7, forward | Lone Star laboratories | 5'-TCCTGAGAGCATCCCTCTAATC-3' | |
| Sequence-based reagent | ATG7, reverse | Lone Star laboratories | 5'- CTTCAGTTCGACACAGGTCATC-3' | |
| Sequence-based reagent | TBP, forward | Lone Star laboratories | 5'-AGTGCCCAGCATCACTGTTT-3' | |
| Sequence-based reagent | TBP, reverse | Lone Star laboratories | 5'-GGTCCATGACTCTCACTTTCTT-3' | |
| Sequence-based reagent | ATG2700 | Dr. Monchaud lab. | ATTCTTGGGGCTGGGGTCCCT TGGGGAACTGTATTGGGTGAACC | |
| Sequence-based reagent | SS-DNA | Dr. Monchaud lab. | GCACGCGTATCTTT TTGGCGCAGGTG | |
| Commercial assay or kit | RNeasy Mini kit | Qiagen | 74104 | |
| Commercial assay or kit | iScript Reverse Transcription SuperMix | BioRad | 1708840 | |
| Chemical compound, drug | Pyridostatin (PDS) | Cayman Chemical | 18013 | |
| Chemical compound, drug | 10-(4'-(N-diethylamino)butyl)−2-chlorophenoxazine (10-NCP) | EMD Millipore | 925681–41 | |
| Chemical compound, drug | N-TASQ | (*Laguerre et al., 2015*; *Laguerre et al., 2016*) | | synthesized by Dr. David Monchaud |
| Software, algorithm | JMP software | SAS Institute, Houston, TX | | |
| Other | Hoechst dye | Santa Cruz Biotechnology | sc-394039 | |
| Other | poly-D-lysine | Millipore | A-003-E | |
| Other | Neurobasal Medium | Life Technologies | 21103–049 | |
| Other | B-27 | Life Technologies | 17504–044 | |
| Other | GlutaMAX | Life Technologies | 35050–061 | |
| Other | penicillin-streptomycin | Life Technologies | 15240.062 | |
| Other | HisPur Ni-NTA resin | Thermo Scientific | 88221 | |
| Other | Dynabeads | ThermoFisher Scientific | 10002D | |

## Chemicals and plasmids

PDS was from Cayman Chemical (#18013). 10-NCP (10-(4'-(N-diethylamino)butyl)−2-chlorophenoxazine) was from EMD Millipore ((#925681–41). Hoechst dye was from Santa Cruz Biotechnology (#sc-394039). N-TASQ was synthesized as described (*Laguerre et al., 2015*; *Laguerre et al., 2016*;

*Yang et al., 2017*). Antibodies against LC3 were from MBL (#PD014). Antibodies against ATG7 (D12B11; #8558), β-actin (8H10D10; #3700), and the Anti-FLAG DYKDDDDK M2 tag (D6W5B; #2368) were from Cell Signaling. Mouse antibodies against MAP2c (A-4, #sc-74421) were from Santa Cruz Biotechnology. Antibodies against Htt (mEM48), rabbit IgG(H+L) conjugated with horseradish peroxidase (HRP) (#AP307P), and mouse IgG(H+L) conjugated with HRP (#AP308P) were from EMD Millipore. Anti-mouse Alexa Fluor 488-labeled (#A11001) and anti-rabbit Alexa Fluor 546-labeled (#A11010) secondary antibodies were from Life Technologies. A single-chain BG4 antibody that recognizes G4 structures (*Biffi et al., 2013*) was purified in the lab of Dr. Nayun Kim. pGW1-Dendra2-LC3 was described (*Tsvetkov et al., 2013b*). pGW1-mito-Keima was cloned from the mt-mKeima/pIND(SP1) construct that was received from Dr. Atsushi Miyawaki (RIKEN Brain Science Institute, Japan). pCAG-TagBFP, pCAG-EGFP-PIF1, pCAG-EGFP-mPIF1 (E307Q *George et al., 2009*), and pEF1A-mApple-ATG7 were cloned by VectorBuilder.

## Cell cultures and transfection

Cortices from rat embryos (E17–18) were dissected, dissociated, and plated on 24-well tissue-culture plates ($4 \times 10^5$/well) coated with poly-D-lysine (BD Biosciences, San Jose, CA), as described (*Moruno-Manchon et al., 2017*; *Moruno Manchon et al., 2015*; *Moruno-Manchon et al., 2018*). Primary cortical neurons were grown in Neurobasal Medium (Life Technologies, Carlsbad, CA) supplemented with B-27 (Life Technologies), GlutaMAX (Life Technologies) and penicillin-streptomycin (Life Technologies). Primary cultures were transfected with Lipofectamine2000 (Thermo Fisher Scientific) and a total of 1–2 μg of plasmid DNA per well, as described (*Moruno-Manchon et al., 2017*; *Moruno Manchon et al., 2015*; *Moruno-Manchon et al., 2018*).

## Survival analysis

We used automated microscopy and longitudinal analysis to determine neuronal survival. This method allows us to track large cellular cohorts and to sensitively measure their survival with the statistical analyses used in clinical medicine (*Moruno Manchon et al., 2015*; *Tsvetkov et al., 2010*; *Arrasate and Finkbeiner, 2005*). For tracking the same group of cells over time, an image of the fiduciary field on the plate was collected at the first time-point and used as a reference image. Each time the same plate was imaged, the fiduciary image was aligned with the reference image. Neurons that died during the imaging interval were assigned a survival time. These events were transformed into log values and plotted in risk of death curves and analyzed for statistical significance (log-rank test). JMP software (SAS Institute, Houston, TX) was used to analyze data and generate survival curves (*Tsvetkov et al., 2013a*; *Tsvetkov et al., 2013b*).

## Optical pulse-chase

Photoswitching of Dendra2-LC3 and Htt$^{ex1}$-Q$_{46}$-Dendra2 was performed as described (*Barmada et al., 2014*; *Tsvetkov et al., 2013b*; *Moruno Manchon et al., 2015*). Upon brief irradiation with short-wave visible light, Dendra2 undergoes an irreversible conformational change ('photoswitch'). The spectral properties of Dendra2 then change from that of a protein that absorbs blue light and emits green fluorescence to that of one that absorbs green light and emits red fluorescence (*Barmada et al., 2014*). Photoswitched Dendra2 maintains these spectral properties until the cell degrades the protein. The red fluorescence intensities from a region of interest in individual cells were measured at different time points. Fluorescence of non-photoswitched 'green' molecules served as a guide for drawing the region of interest. The decays of red fluorescence were plotted against time, transformed into log values, and individual half-life (t$_{1/2}$) was analyzed (*Barmada et al., 2014*; *Tsvetkov et al., 2013b*). The half-lives (H$^{1/2}$) of Dendra2-LC3 was calculated using the formula: $H^{1/2} = (24 \times Ln(2))/(Ln(A/A°)$. A = final fluorescence; A°=initial fluorescence.

## Immunoblotting

Neuronal cultures were lysed in RIPA buffer (150 mM NaCl, 1% Nonidet P40, 0.5% sodium deoxycholate, 0.1% SDS and 50 mM Tris/HCl (pH 8.0), with phosphatase and protease inhibitors cocktail) on ice. Lysates were vortexed and cleared by centrifugation (14000 g, 10 min, 4°C). Supernatants were collected, and protein concentrations were determined by the Bicinchoninic Acid Protein Assay Kit (Thermo Scientific). Samples were analyzed by SDS/PAGE (4–12% gradient gels), and proteins

were transferred on to nitrocellulose membranes using the iBlot2 system (Life Technologies). Membranes were blocked with 5% skimmed milk for 1 hr at room temperature, and they were incubated with the primary antibodies (anti-LC3, Htt, anti-actin or anti-ATG7) overnight at 4°C. Membranes were then washed with TBS (Tris-buffered saline; 10 mM Tris/HCl and 150 mM NaCl (pH 7.4)) and incubated with anti-rabbit-HRP or anti-mouse-HRP for 1 hr at room temperature. Chemiluminescent signal was visualized with Prometheus ProSignal Pico (Genesee Scientific) on Blue Devil autoradiography films (Genesee Scientific).

## G4-DNA analyses

The QGRS mapper (http://bioinformatics.ramapo.edu/QGRS/index.php) was used to determine the potential G4-DNA structures contained in genes of interest and their G-scores. Search parameters: maximal length: 45; minimal G-group size: 3; loop size: from 0 to $10^2$.

## RNA extraction and qRT-PCR

Total RNA was extracted from primary culture using the RNeasy Mini kit (#74104, Qiagen), and then reverse transcribed using iScript Reverse Transcription SuperMix (#1708840, Bio-Rad), according to the manufacturer's protocol and as described (*Moruno-Manchon et al., 2017*). RT-qPCR was performed using a Bio-Rad CFX96 Touch machine using SSoAdvanced Universal SYBR Green (#1725275, Bio-Rad) for visualization and quantification according to the manufacturer's instructions. Primer sequences were: ATG7 (*Atg7*), forward: 5′-TCCTGAGAGCATCCCTCTAATC-3′, reverse: 5′- C TTCAGTTCGACACAGGTCATC-3′; TBP (*Tbp*), forward: 5′-AGTGCCCAGCATCACTGTTT-3′, reverse: 5′-GGTCCATGACTCTCACTTTCTT-3′. The PCR conditions were: 95°C for 3 min, followed by 40 cycles of 95°C for 10 s and 55°C for 30 s. Relative expression levels were calculated from the average threshold cycle number using the delta-delta Ct method.

## Oligonucleotides

The sequences of oligonucleotides used herein were: Atg7-32, 5′-GGGGCTGGGGTCCC TTGGGGAACTGTATTGGG-3′; mutAtg7-32, 5′-GCGCCTGCGCTCCCTTGCGCAACTGTATTGCG-3′; fam-Atg7-32-tamra, 5′-*fam*-GGGGCTGGGGTCCCTTGGGGAACTGTATTGGG-*tamra*-3′; fam-mutAtg7-32-tamra: 5′-*fam*-GCGCCTGCGCTCCCTTGCGCAACTGTATTGCG-*tamra*-3′. The lyophilized DNA strands purchased from Eurogentec (Seraing, Belgium) were firstly diluted at 500 μM in deionized water (18.2 MΩ.cm resistivity). DNA samples were prepared in a Caco.K10 buffer, composed of 10 mM lithium cacodylate buffer (pH 7.2) plus 10 mM KCl/90 mM LiCl. Samples were prepared by mixing 40 μL of the constitutive strand (500 μM) with 8 μL of a lithium cacodylate buffer solution (100 mM, pH 7.2), plus 8 μL of a KCl/LiCl solution (100 mM/900 mM) and 24 μL of water. The actual concentration of each sample was determined through a dilution to 1 μM theoretical concentration *via* a UV spectral analysis at 260 nm (after 5 min at 90°C) with the following molar extinction coefficient (ε) values: 302000 (Atg7-32), 276500 (mutAtg7-32), 355300 (fam-Atg7-32-tamra) and 329800 l.mol$^{-1}$.cm$^{-1}$ (fam-mutAtg7-32-tamra). The G4 structures were folded heating the solutions at 90°C for 5 min, and then cooling them on ice (for 7 hr) before being stored overnight at 4°C.

## CD and TDS experiments

CD and UV-Vis spectra were recorded on the JASCO J-815 spectropolarimeter and the JASCO V630Bio spectrophotometer, respectively, in a 10 mm path-length quartz semi-micro cuvette (Starna). CD spectra of 3 μM of Atg7-32 and mutAtg7-32 (Eurogentec) were recorded over a range of 220–340 nm (bandwidth = 1 nm, 1 nm data pitch, 1 s response, scan speed = 500 nm.min$^{-1}$, averaged over five scans) without and with dehydrating agent (PEG200, 20% v/v; acetonitrile, 50% v/v) in 600 μL (final volume) of in 10 mM lithium cacodylate buffer (pH 7.2) plus 10 mM KCl and 90 mM LiCl (Caco.K10). Final data were treated with OriginPro8, zeroing CD spectra at 340 nm. TDS experiments were performed with Atg7-32 and mutAtg7-32 (3 μM) recording the optical over a range of 220–340 nm at 20°C and 80°C in 600 μL (final volume) of Caco.K10. Final data were treated with Excel (Microsoft Corp.) and OriginPro9.1 (OriginLab Corp.). TDS signature were calculated subtracting the spectra collected at 20°C from the spectra collected at 80°C, normalized (0 to 1) and zeroed at 340 nm.

## FRET-melting experiments

Experiments were performed in a 96-well format using a Mx3005P qPCR machine (Agilent) equipped with FAM filters ($\lambda_{ex}$ = 492 nm; $\lambda_{em}$ = 516 nm) in 100 µL (final volume) of Caco.K10 with 0.2 µM of Fam-Atg7-32-Tamra or Fam-mutAtg7-32-Tamra (Eurogentec) with 0, 1, 2 and 5 molar equivalents of PDS and BRACO-19 (*i.e.*, 0, 0.2, 0.4 and 1.0 µM ligand). After a first equilibration step (25°C, 30 s), a stepwise increase of 1°C every 30 s for 65 cycles to reach 90°C was performed, and measurements were made after each cycle. Final data were analyzed with Excel (Microsoft Corp.) and OriginPro9.1 (OriginLab Corp.). The emission of FAM was normalized (0 to 1), and $T_{1/2}$ was defined as the temperature for which the normalized emission is 0.5; $\Delta T_{1/2}$ values, calculated as follows: $\Delta T_{1/2}$ = [$T_{1/2}$(DNA+ligand)-($T_{1/2}$(DNA alone)], and are means of three experiments.

## NMR experiments

Atg7-32 and mutAtg7-32 (Eurogentec) were annealed at 200 µM in a Caco.K10 by heating at 95°C for 10 min. The samples were cooled to 4°C (ice bath) and equilibrated at 4°C for at least 24 hr. $^1$H-NMR spectra (250 µL final volume) were acquired after the addition of DSS (4,4-dimethyl-4-silapentane-1-sulfonic acid) as internal calibration standard. NMR spectra were recorded at 298 K (4248 scans) using a 600 MHz Bruker Avance III HD spectrometer equipped with a cryogenic probe. Water suppression was achieved using excitation sculpting (pulse program: zgesgp). Final data were analyzed with TopSpin v4.0.6 (Bruker).

## HF2 binding assay

HF2 antibody expression and purification were carried out as described (*Fernando et al., 2008*). The expression of the HF2 single–chain antibody was then induced by 100 mM isopropyl β-D-1-thiogalactopyranosid in *E. coli*. Cells were pelleted and resuspended in a lysis buffer (25 mM Tris-HCl, 100 mM NaCl, 10% glycerol, 1% NP-40 and 10 mM imidazole) and sonicated using the QSONICA sonicator. Purification of the 6XHis-tagged HF2 antibody was carried out using HisPur Ni-NTA resin according to the manufacturer's instruction (Thermo Scientific). The eluted protein was concentrated with the Amicon Ultra-4 Centrifugal Filter and stored at −20°C in 50% glycerol. For the binding assay, 5′- Cy5-labeled oligonucleotides (Sigma) were resuspended in 10 mM Tris-Cl containing 100 mM LiCl or KCl and denatured at 95°C for 5 min and then slowly cooled overnight to allow secondary structure formation. Annealed oligonucleotides were mixed with the purified HF2 antibody in 100 mM LiCl or KCl, 20 mM HEPES pH 7.5, 0.01% NP40, 5% glycerol, 5 mM MgCl$_2$, and incubated at room temperature for 15 min before running on a 10% non-denaturing TBE-polyacrylamide gel with 0.5X TBE. Gel images were captured using the BioRad Chemidoc imager. Sequences of the oligonucleotides were:

    ATG2700, ATTCTTGGGGCTGGGGTCCCTTGGGGAACTGTATTGGGTGAACC
    SS-DNA, GCACGCGTATCTTTTTGGCGCAGGTG

## DNA-dynabeads affinity purification of proteins

DNA-Dynabeads affinity purification of proteins was carried out as described (*Gao et al., 2015*) with several modifications. For DNA-conjugated Dynabeads preparation, biotinylated oligonucleotides ATG-2700 and SS-DNA were ordered from Sigma. The oligonucleotides were incubated at 60°C overnight in the presence of 10 mM Tris pH 7.5 and 100 mM KCl and then conjugated to Streptavidin-Coupled M-280 Dynabeads (Life Technologies) as per the manufacturer's instructions. Yeast extract was made by glass bead-mediated cell disruption in 2 ml of lysis buffer (50 mM HEPES–NaOH, pH 7.5, 300 mM KCl, 1 mM EDTA, 10% glycerol, 0.05% NP-40, 1 mM DTT, 1 mM PMSF, 1X protease inhibitor cocktail (Roche)). After mechanical lysis of cells with Biospec Mini-bead-beater, the cell lysate was collected in a 15 ml tube and sonicated. DNA-conjugated Dynabeads were washed once with the lysis buffer and incubated overnight with gentle inversion with the remaining yeast extract at 4°C. The beads were washed with the lysis buffer five times and then eluted by boiling in 1XSDS-PAGE loading buffer followed by immunoblotting analysis with the anti-FLAG (Sigma; # A8592) or anti-His (Sigma; # H1029) antibodies.

## Fluorescence microscopy

Live cell and fixed cell imaging was performed with the EVOS FL Auto Imaging System (Thermo Fisher Scientific). Lipofuscin was measured in the brain samples acquired from the aged female and male mice treated with a vehicle or with PDS (5 mg/kg). Brain samples were mounted on the glass slides and stained with the nuclear Hoechst dye. Samples were then imaged using the green GFP filter for autofluorescent lipofuscin and the blue DAPI filter with the EVOS microscopy system.

## Immunocytochemistry

Cultured primary cortical neurons on coverslips were treated with a vehicle or with PDS overnight, fixed with 4% paraformaldehyde, permeabilized with a 0.5% Triton X-100/PBS solution, and blocked with a 5% bovine serum albumin/PBS solution. Neurons were then stained with antibodies against MAP2c and with the G4-selective fluorophore N-TASQ overnight. Neurons were incubated with secondary antibodies, stained with Hoechst dye, and imaged with the EVOS microscopy system.

## Immunohistochemistry

To determine and analyze G4 quadruples in brain samples from young and aged mice, frozen floating brain sections were incubated with antibodies against G4 (BG4) overnight. Samples were then incubated with antibodies against FLAG for 1 hr at room temperature, and then with secondary antibodies conjugated with a fluorochrome for 1 hr at room temperature. Nuclei were stained with Hoechst dye. Brain sections were mounted on glass slides, and imaged with a Leica DM8i SPE confocal microscope or the EVOS microscope.

## Novel object recognition test (NORT)

The test is a standard test for recognition memory that is sensitive to aging. For this test, we used old male and female mice (25 months). During the test, two identical objects were presented to each mouse in an arena, and the mice allowed to explore the objects for 10 min. The objects and their position in the arena will be pseudo-randomized between the different mice. After 1 hr interval, one of these objects was replaced with a novel object; again, the mouse was allowed to explore the objects for 10 min. We video recorded the behavior of mice and evaluated the differences in the exploration time with novel and familiar objects. The discrimination index (DI) was calculated as: $DI = (TN*100)/(TN+TF)$. TN is the time mouse spent exploring the novel object. TF is the time mouse spent exploring the familiar object. All test were performed by an investigator blinded to treatment group.

## Statistical analysis

For longitudinal survival analysis, neurons that died during the imaging interval were assigned a survival time (the period between transfection and their disappearance from an image). These event times were used to generate exponential cumulative survival curves in JMP statistical software. Survival curves describe the risk of death for single cells in the group being longitudinally imaged. To determine differences in the survival curves, they were then analyzed for statistical significance by the log-rank test as described (*Moruno-Manchon et al., 2017*; *Moruno Manchon et al., 2015*; *Moruno-Manchon et al., 2018*). To compare differences across two groups, the groups were analyzed with Student's t-test. Differences across multiple groups were analyzed with one-way ANOVA.

## Acknowledgements

We thank members of the AST and LDM laboratories for useful discussions. Raquel Cornell, Sharon Gordon, Summer Hensley, Diana Parker, and Martha Belmares provided administrative assistance.

## Additional information

### Funding

| Funder | Grant reference number | Author |
|--------|------------------------|--------|
| National Institute of General Medical Sciences | GM116007 | Nayun Kim |
| Welch Foundation | AU1875 | Nayun Kim |
| Agence Nationale de la Recherche | ANR-17-CE17-0010-01 | David Monchaud |
| National Institute of Neurological Disorders and Stroke | R01NS094543 | Louise D McCullough |

The funders had no role in study design, data collection and interpretation, or the decision to submit the work for publication.

### Author contributions

Jose F Moruno-Manchon, Formal analysis, Investigation, Methodology; Pauline Lejault, Yaoxuan Wang, Brenna McCauley, Pedram Honarpisheh, Diego A Morales Scheihing, Shivani Singh, Data curation, Formal analysis, Investigation, Methodology; Weiwei Dang, Nayun Kim, Resources, Data curation, Formal analysis; Akihiko Urayama, Resources, Data curation, Formal analysis, Investigation; Liang Zhu, Formal analysis, Writing - review and editing; David Monchaud, Conceptualization, Resources, Data curation, Formal analysis, Supervision, Investigation; Louise D McCullough, Conceptualization, Resources, Supervision; Andrey S Tsvetkov, Conceptualization, Data curation, Supervision, Project administration

### Author ORCIDs

Jose F Moruno-Manchon ORCID https://orcid.org/0000-0002-2139-6134
Pedram Honarpisheh ORCID http://orcid.org/0000-0002-9126-6271
Weiwei Dang ORCID http://orcid.org/0000-0002-6931-4636
David Monchaud ORCID http://orcid.org/0000-0002-3056-9295
Louise D McCullough ORCID http://orcid.org/0000-0002-8050-1686
Andrey S Tsvetkov ORCID https://orcid.org/0000-0001-9749-9618

### Ethics

Animal experimentation: This study was performed in strict accordance with the recommendations in the Guide for the Care and Use of Laboratory Animals of the National Institutes of Health. All of the animals were handled according to approved institutional animal care and use committee (IACUC) protocols (#AWC-16-0081) of the University of Texas. The protocol was approved by the Committee on the Ethics of Animal Experiments of the University of Texas.

### Decision letter and Author response

Decision letter https://doi.org/10.7554/eLife.52283.sa1
Author response https://doi.org/10.7554/eLife.52283.sa2

## Additional files

### Supplementary files

• Transparent reporting form

### Data availability

All data generated or analysed during this study are included in the manuscript and supporting files.

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
