## [Decision Letter]

**Decision letter after peer review:**

[Editors’ note: the authors submitted for reconsideration following the decision after peer review. What follows is the decision letter after the first round of review.]

Thank you for submitting your work entitled "A small molecule G-quadruplex stabilizer reveals a novel pathway of autophagy regulation in neurons" for consideration by *eLife*. Your article has been reviewed by a Senior Editor, a Reviewing Editor, and three reviewers. The reviewers have opted to remain anonymous.

Our decision has been reached after consultation between the reviewers. Based on these discussions and the individual reviews below, we regret to inform you that your work will not be considered at this moment for publication in *eLife*. However, we believe that this is a very interesting study that after further work would be appropriate for *eLife*. It reveals that G4 may represent a new putative intervention point to interfere with autophagy-related neurodegeneration represents an important discovery and that the implication of G4-DNA in autophagy and age-related neurological deficit is novel and will be of interest to a broad readership and of interest for *eLife*. We would invite to resubmit a properly revised version attending the comments of the three reviewers accompanying your submission with a rebuttal explaining point-by-point the comments of the reviewers.

Reviewer #1:

Summary:

The G-quadruplex (G4) ligand pyridostatin (PDS) was found to downregulate expression of the Atg7 gene in neurons. The first intron of the Atg7 gene contains predicted G4-forming sequence that was indeed shown to form G4 and interact with PDS. Consistent with these findings, in vitro a G4 antibody and G4-binding protein bind to the Atg7 intron G4-sequence. In vivo, mice treated with PDS were found to develop memory deficits and accumulation of lipofuscin, a mixture of oxidized lipids and proteins previously observed to accumulate in aged brains. Brain samples from aged mice, but not young mice, contained G4 DNA, as evidenced by staining with G4-selective reagent. Overexpression of the G4-resolving helicase Pif1 in neurons improved phenotypes associated with PDS treatment, i.e., neuronal death. Based on their findings, the authors conclude that G4 DNA is involved in regulating autophagy in neurons.

Critical Comments:

- The narrative of the Results section is noticeably deficient in making quantitative statements relating to the experimental data. This issue should be addressed throughout the Results section.

- A useful control in the Pif1 rescue experiments would have been to test if a site-directed ATPase/helicase-dead mutant version of Pif1 failed to affect PDS-induced phenotypes in cultured primary neurons.

- Please define/explain Dendra2-LC3 half-life and how it is a reliable indicator of autophagy. It is mentioned in Figure 4 that LC3 is an autophagy marker. This should be better described in Results section and reference(s) provided.

- Aside from short-term memory as assayed by novel object recognition test, are there any other effects on mice treated with PDS in terms of development of memory deficits or neurological function/capacity? Were there any sex-specific effects?

- The authors briefly mention in the Introduction the recently published paper by Beauvarlet et al., (April 2019) suggesting that G4 DNA in autophagic genes regulates autophagy in cancer cells. As cite in that work, there were several other previously published papers suggesting a connection of autophagy to G4 nucleic acid metabolism as probed by G4 ligands: Orlotti et al., (2012); Zhou et al., (2009); Zhou et al., (2009). While all these papers delved more into the relationships of G4 and autophagy in cancer, it would be useful to discuss these works in a single paragraph in the Discussion and place the current work in light of those findings.

- In the Introduction, references should be provided for the statement that G4 has been implicated in neurodegenerative disorders frontotemporal dementia and ALS.

- In previous work by the authors, they reported that PDS promotes DNA damage and downregulates transcription of BRCA1 in neurons (Aging (2017)). The showed that overexpressed BRCA1 mitigates PDS-induced DNA damage. Similarly, in the current work does overexpression of ATG7 enzyme modulate the PDS-related phenotypes observed? Is DNA damage accumulation in neurons also observed in the current work? How are the results and findings from the two studies related, if at all?

Reviewer #2:

In this paper, the authors rather convincingly show that the ATG7 gene, which is critical for the initiation of autophagy and whose transcription decreases with aging, does contain a bona fide G-quadruplex (G4) in its first intron and that stabilization of G4 using pyridostatin (PDS), a well-known benchmark G4 ligand, downregulates this ATG7 gene. All this suggest that stabilization of ATG7 G4 does interfere with the transcription of this gene, thereby inhibiting induction of autophagy. In good agreement, the authors found that mice treated with PDS develop memory deficits and accumulation of lipofuscin, suggesting premature aging and deficient autophagy. Moreover, brain samples from aged mice contain G4-DNA which are absent in brain samples from young mice. Finally, the authors showed that overexpressing the helicase Pif1, which is known to resolve G4, in neurons exposed to PDS improves the various phenotypes associated with PDS treatment, thereby suggesting that G4-DNA may represent an interesting and relevant intervention point for boosting autophagy and thereby interfering with neurodegeneration.

In general, I think this is a very interesting study based on well-designed and well-conducted experiments that deserves to be published in *eLife*. In particular, revealing that G4 may represent a new putative intervention point to interfere with autophagy-related neurodegeneration represents an important discovery. Also, the idea that G4-ligands may induce/accelerate neurodegeneration is highly important, especially with respect to their putative development to the clinics in other field like cancerology. However, I have a few comments that need to be addressed before this paper could be deemed for publication in *eLife*.

- Figure 1: the search for PQFS in the gene and promoter sequences of autophagy genes. The analysis performed by the authors indicates that all the autophagy genes contain putative G4-DNA. It would be important to precise the percentage of total genes that contain, or not putative G4-DNA to assess how specific is this correlation.

- Figure 2: 2 µM PDS has a much stronger negative effect than 2 µM BRACO19 on the level of the ATG7 transcript. However, the effect on ATG7 protein level is similar for both compounds suggesting that BRACO19 could also interfere with translation of ATG7 mRNA or on ATG7 protein stability. The authors should comment on this. As for BRACO19, although it is very important to use another GA-ligand than PDS to support the authors' main conclusions, it is somehow surprising that BRACO19 is used only in this experiment and also that its putative stabilizing effect on ATG7 G4-DNA is not properly assessed (see below).

- Figure 3: The CD signature of the ATG7 gene in the classical conditions is not fully convincing and could correspond to a mix of several structures (not only G4). The authors argue that the presence of dehydrating agents (PEG, CH_3_CN) reduces the polymorphism and indeed the CD looks better but this effect has been reported only for telomeric sequences and is not fully admitted. Therefore, the salt effect (Li^+^ to K^+^) should preferably been tested using the HF2 antibodies. In line with what I discussed just above, BRACO19 should also be tested in the UV-melting experiments performed in panel c to determine if it also stabilizes, or not, the most probable ATG7 G4 and also in the experiments performed with N-TASQ on neurons in panels I & J. The TDS spectrum without dehydrating agents should also be shown. TDS and CD spectra should be shown on different panels and the CD in K^+^ conditions should not be termed "cont" (for control?) as it is misleading. Also, another classical control for CD which is missing is the use of scrambled G runs. Another important point is that the in vitro experiments presented in panels e to h should be repeated in presence, or not, of PDS or of BRACO19 to determine if these two G4 ligands may, or not, interfere with the binding of the HF2 antibody and/or of G4-binding proteins. This point looks important to me as the general assumption is that G4-ligands stabilize G4 structures but, in principle they could also destabilize them, and/or even prevent the binding of various factors such as antibodies or G4-binding proteins by direct competition. Finally, as for the experiments with N-TASQ, I have some problem to understand them because, as this fluorescent molecule is also a G4-ligand, one may imagine a competition between this compound and PDS for the binding on G4. And, indeed, in the original paper on N-TASQ (Laguerre et al., 2016), this problem is discussed and addressed (by using BRACO19 instead of PDS) and the authors concluded that at high concentration (100 µM) N-TASQ provides high resolution images but does not allow to visualize significant differences between BRACO19-treated and -untreated cells and that the only conditions that allowed to visualize an increase in N-TASQ stained nuclear foci was a low dose of N-TASQ (2.5 µM) and of BRACO19, and that a higher dose of BRACO19 leads to a BRACO19 dose-dependent decrease in N-TASQ staining, presumably because of a competition between these two G4-ligands for binding on G4. Hence it is hard for me to understand why and how PDS treatment should lead to an increase in N-TASQ staining (used at 50 µM here) as shown in panel i and j. Of note such a competition with PDS has been described for DAOTA-M2, another G4-specific fluorescent probe (Shivalingam et al., 2015). All this should be discussed and the effect of BRACO19 on N-TASQ staining should also be tested.

- Figure 4: BRACO19 should also be tested in at least one of the experiments presented here as there are at the basis of the main conclusion of the paper, destabilization of DNA-G4 represents a relevant and interesting intervention point ATG7 to interfere with neurodegeneration associated with aging-related decline in induction of autophagy.

- In the Discussion section, my suggestion is that the authors should discuss about the ability of G4-ligand to stabilize DNA-G4. Is it a general property of all the G4-ligands or is it specific to a subset of G4-ligands (that includes PDS and BRACO19)? Should a compound that efficiently bind G4 without any effect on their stability exist, then it would represent an ideal control to further validate their findings. Also the possibility that the PDS-related phenotypes may involve its effect on RNA-G4 should be mentioned and discussed.

To finish, and importantly, in my view the main message of this manuscript is that G4-DNA may represent an interesting and relevant intervention point for boosting autophagy and thereby interfering with neurodegeneration, rather than the discovery of a novel pathway that regulates autophagy in neurons, as stated by the authors already in the title. Indeed, if the authors want to state that their findings do reveal a novel pathway for regulating autophagy, than they need to find physiological situations where the stability of DNA-G4 present in autophagy genes (in particular in ATG7) may be tuned by various cellular pathway(s)/component(s) which, this way, regulate autophagy. As for now, they essentially showed that stabilizing G4 using PDS downregulates ATG7 and that PDS induces memory deficits and autophagy in mice and that, on the contrary, overexpressing the G4-DNA helicase Pif1 in neurons exposed to PDS suppresses PDS-associated phentotype. Not to mention that this G4 stabilization/PDS effect may also be at the level of G4-RNA. Therefore, I suggest that the authors down tune their message, especially in the Title but also in the discussion. I guess that revealing a new and relevant intervention point for modulating autophagy in neurons is per se sufficiently interesting in addition to be of biomedical relevance.

Reviewer #3:

The authors used the G4 stabilizer pyridostatin (PDS) in cultured neurons and in mice to unravel a role for G4-DNA structures during Atg7-mediated autophagy. They used biophysical methods to demonstrate accumulation of G4-DNA in the Atg7 gene and immunostaining methods to illustrate an age-dependent global increase of G4-DNA. The authors further show that the majority of phenotypes induced by PDS were partially rescued by overexpression of fhe Pif1 helicase.

The manuscript is clearly written, and data are well presented. The implication of G4-DNA in autophagy and age-related neurological deficit is novel and will be of interest to a broad readership. It is not, however, clear if the effects reported here are direct consequences of G4-DNA in the Atg7 gene or indirect global perturbations of G4-DNA homeostasis. PDS is a blunt tool potentially impacting over 600,000 putative G4-DNA structures. Also, Pif1, as acknowledged by the authors, impacts telomere length, so how can the authors be sure that the reported partial improvements of neuronal phenotypes are solely due to resolving G4-DNA structures? Is the partial rescue specific to G4s in the Atfg7 gene? Performing a classical epistasis experiments is critical in this manuscript. For example, repeating key experiments in presence and absence of Atg7 will confirm that the reported phenotypes are due to direct modulation of G4-DNA in the Atg7 gene, hence supports the conclusion of a novel pathway as stated in the Title.

For the huntingtin experiments in Figure 4, I suggest using patient derived fibroblasts or iPS-derived striatal neurons instead of ectopic expression of the exon-1 fragment of the poly-Q huntingtin. Although, in silico predictions using the QGRS mapper rule out G4-DNA, it is important to experimentally rule it out in the native genomic environment.

The protein p62 is a known hallmark of perturbed autophagy. Is p62 aggregation also modulated by PDS in an Atg7 dependent manner? This is important since this hallmark protein aggregation is a common phenomenon in a number of age-associated neurological disorders including Huntington's and ALS/FTD. In the opinion of this reviewer, this is a better hallmark of perturbed autophagy given its clinical relevance.

Overall, the concept is novel and exciting but the data in its present form do not support the main conclusion.

[Editors’ note: further revisions were suggested prior to acceptance, as described below.]

Thank you for submitting your article "Small-molecule G-quadruplex stabilizers reveal a novel pathway of autophagy regulation in neurons" for consideration by *eLife*. Your article has been reviewed by Michael Eisen as the Senior Editor, a Reviewing Editor, and two reviewers. The following individuals involved in review of your submission have agreed to reveal their identity: Sherif El-Khamisy (Reviewer #1).

The reviewers have discussed the reviews with one another and the Reviewing Editor has drafted this decision to help you prepare a revised submission.

Summary:

This manuscript is a resubmission of a previous one in which authors show that the G-quadruplex (G4) ligand pyridostatin (PDS) was found to downregulate expression of the Atg7 gene in neurons. The first intron of the Atg7 gene contains predicted G4-forming sequences that seem to form G4 and interact with PDS. Mice treated with PDS develop memory deficits and accumulation of lipids and proteins previously observed to accumulate in aged brains. Brain samples from aged mice, but not young mice, contained G4 DNA, and overexpression of the G4-resolving helicase Pif1 in neurons improved the phenotypes associated with PDS treatment. Based on their findings, the authors conclude that G4 DNA is involved in regulating autophagy in neurons. The authors have satisfactorily responded to the concerns raised by the referees, but q few points need to be taken before the manuscript can be accepted.

Essential revisions:

- The 3 quartet structure shown in Figure 3 has a low probability of formation due to the presence of 3 long loops (5-nt, 7-nt, 9-nt) which drastically reduce its stability (see various methods of the G4 score calculation in Bedrat et al., 2016; Puig-Lombardi et al., 2019). Hence it follows that the Atg7-32 sequence is most probably highly dynamic and may form several secondary structures that exist in equilibrium (various G4, hairpins etc.), which is good agreement with the very broad profile of the NMR spectra. Hence this analysis does not allow the authors to firmly conclude the existence of a stable G4. Therefore, the authors should down-tune, or at least modulate their G4 hypothesis.

- The fact that Pif1 rescues PDS-induced phenotypes in cultured primary neurons (Figure 8). This observation is interesting but somehow a bit surprising and rather counter-intuitive as it is not fully consistent with numerous studies reported in the literature that show that the G4 unwinding activity of most of the G4 helicases is indeed prevented by G4 ligands. This has been shown in particular for Pif1 (see Mendoza et al., 2016; Mergny et al., 2015; Balasubramanian et al., 2015 plus references cited therein). Therefore, the assumption that Pif1 rescues PDS-induced phenotype by unwinding G4 in the Atg7 is unclear. The authors should discuss the results of their experiment with Pif1 in light of all the published data indicating that Pif1 G4 unwinding activity is inhibited by various G4 ligands that include PDS, or alternatively they could test their hypothesis by performing a functional in vitro assay (e.g.: unwinding assay with or without PDS).

---

## [Author Response]

[Editors’ note: the authors resubmitted a revised version of the paper for consideration. What follows is the authors’ response to the first round of review.]

Reviewer #1:Summary:The G-quadruplex (G4) ligand pyridostatin (PDS) was found to downregulate expression of the Atg7 gene in neurons. The first intron of the Atg7 gene contains predicted G4-forming sequence that was indeed shown to form G4 and interact with PDS. Consistent with these findings, in vitro a G4 antibody and G4-binding protein bind to the Atg7 intron G4-sequence. In vivo, mice treated with PDS were found to develop memory deficits and accumulation of lipofuscin, a mixture of oxidized lipids and proteins previously observed to accumulate in aged brains. Brain samples from aged mice, but not young mice, contained G4 DNA, as evidenced by staining with G4-selective reagent. Overexpression of the G4-resolving helicase Pif1 in neurons improved phenotypes associated with PDS treatment, i.e., neuronal death. Based on their findings, the authors conclude that G4 DNA is involved in regulating autophagy in neurons.Critical Comments:- The narrative of the Results section is noticeably deficient in making quantitative statements relating to the experimental data. This issue should be addressed throughout the Results section.

We thank the reviewer for this comment. We revised the manuscript to describe the results in greater detail, including quantitative statements for all of the results in the revised Results section.

- A useful control in the Pif1 rescue experiments would have been to test if a site-directed ATPase/helicase-dead mutant version of Pif1 failed to affect PDS-induced phenotypes in cultured primary neurons.

We thought this was an excellent suggestion, and we collected new data, which were added to the revised manuscript, to address this issue (Figure 8F). We cloned a site-directed ATPase/helicase-dead mutant version of Pif1 (E307Q) (George et al., 2009) and expressed it in primary neurons. Indeed, the mutant Pif1 failed to affect PDS-associated neuronal phenotypes—strong evidence that wild-type Pif1 activates coping mechanisms in PDS-treated degenerating neurons, leading to improved neuronal phenotypes, such as enhanced autophagy.

- Please define/explain Dendra2-LC3 half-life and how it is a reliable indicator of autophagy. It is mentioned in Figure 4 that LC3 is an autophagy marker. This should be better described in Results section and reference(s) provided.

The referee correctly points out that we only briefly described how protein (or organelle) half-life is measured with photoswitchable proteins and optical pulse-chase labeling (OPL). A number of studies used Dendra2 or other photoswitchable proteins, such as EOS2, and the OPL method to study protein or organelle dynamics in live cells. For example, mouse cell lines that express a mitochondrially localized Dendra2 (mito-Dendra2) were created to study mitochondrial dynamics and mitophagy (Pham, McCaffery and Chan, 2012). Another group created Dendra2-based PhOTO zebrafish for studying development and regeneration (Dempsey, Fraser and Pantazis, 2012). Dendra2-based OPL has been applied to study autophagy with Dendra2LC3 (Barmada et al., 2014; Moruno Manchon et al., 2015; Moruno Manchon et al., 2016); Tsvetkov et al., 2013) (LC3 is a marker of autophagy (Klionsky et al., 2016; Mizushima, Yoshimori and Levine, 2010), protein degradation (Barmada et al., 2014; Fernando, Rodriguez and Balasubramanian, 2008; Kwok and Merrick, 2017; Tsvetkov et al., 2013; Skibinski et al., 2017), and the dynamics of synaptic proteins (Wang et al., 2009). The journal Autophagy published a review paper, which recommended using Dendra2-LC3 to study autophagic flux in neurons (Klionsky et al., 2016). To measure autophagic flux in live neurons, we used the OPL method and longitudinal imaging (Barmada et al., 2014; Tsvetkov et al., 2013). Brief irradiation with short wavelength visible light irreversibly changes the conformation of “green” Dendra2 (“photoswitch”) and its fluorescence to red. We can then track how the red signal (e.g., “red” Dendra2-LC3) is “cleared” over time and measure the half-life of Dendra2-LC3. This information and references have been added to the manuscript, thereby directly addressing the reviewer’s concern.

- Aside from short-term memory as assayed by novel object recognition test, are there any other effects on mice treated with PDS in terms of development of memory deficits or neurological function/capacity? Were there any sex-specific effects?

We thank the reviewer for these questions. It took nearly 6 months to execute the requested experiments and revise the manuscript, primarily because the mice needed to reach a certain age to perform the requested task assays (we age our animals in house to control for diet/housing/microbiome effects). We apologize for this delay.

Regarding sex differences, we only used male mice in the original submission. This is a very important point and thus we prepared new cohorts of male and female mice that were injected with PDS. Previously, a G4-binding small molecule (MM41) was used as an anti-cancer therapy (Kulkarni, Chen and Maday, 2018); we therefore used a comparable dosage and schedule of PDS. We also used old male and female mice (25 months) with a leaky blood brain barrier (BBB) (Haeusle, Donnelly and Rothstein, 2016). All test were performed by an investigator blinded to treatment group. We examined *these* mice in (1) the novel object recognition test (NORT), (2) a fear conditioning test. We worked in collaboration with Dr. McCullough, who routinely uses animal models of stroke and has extensive expertise with behavior studies (https://med.uth.edu/neurology/faculty/louise-d-mccullough-md-phd/). First, we found that male and female mice injected with PDS performed significantly worse in the NORT (Figure 6A; *p-value(male-cont vs PDS)=0.0265, *pvalue(female-cont vs PDS)=0.0382, p-value(male vs female)=0.1029 (two-way ANOVA)). Second, mice were tested in the fear conditioning assay. The latter proved *extremely stressful* for old mice: they would frequently not move, making it difficult to analyze the data. Nevertheless, male mice performed worse in the fear conditioning assay. There was also a trend showing worse performance in the female cohort, although the data did not achieve statistical significance (see Author response image 1). Due to the lack of mobility in the aged mice (which could be mistaken for “freezing”), we did not include these data into the manuscript.

**Author response image 1. respfig1:** Two-way ANOVA, p value for control/PDS main effect is significant, p=0.0203, and the effects on male vs female is also significant, p=0.0326.

Sidak’s multiple comparisons test shows that there is a significant difference between control and PDS within the male group after adjustment for multiple testing (p=0.0372), but not within the female group.

- The authors briefly mention in the Introduction the recently published paper by Beauvarlet et al., (April 2019) suggesting that G4 DNA in autophagic genes regulates autophagy in cancer cells. As cite in that work, there were several other previously published papers suggesting a connection of autophagy to G4 nucleic acid metabolism as probed by G4 ligands: Orlotti et al., (2012); Zhou et al., (2009); Zhou et al., (2009). While all these papers delved more into the relationships of G4 and autophagy in cancer, it would be useful to discuss these works in a single paragraph in the Discussion and place the current work in light of those findings.

We thank the reviewer for this comment. That is an excellent suggestion, as we indeed only briefly mentioned that a prior study investigated whether there is a relationship between G4-DNA, autophagy, and cancer. Critically, we did not emphasize that G4 ligands *stimulate* autophagy in cancer cells. In our study, we show the opposite: G4-DNA ligands downregulate autophagy in post-mitotic neurons, which comes as no surprise since the autophagic pathways in neurons differ from those in other cell types (Kulkarni, Chen and Maday, 2018).Therefore, we have taken this opportunity to review our discussion of the literature and to revise the manuscript to more effectively elaborate on the involvement of G4-DNA in autophagy and, importantly, on potential differences between neurons and cancer cells.

- In the Introduction, references should be provided for the statement that G4 has been implicated in neurodegenerative disorders frontotemporal dementia and ALS.

We added a reference to illustrate that the G4 structures play a role in frontotemporal dementia and ALS (Haeusler, Donnelly and Rothstein, 2016).

- In previous work by the authors, they reported that PDS promotes DNA damage and downregulates transcription of BRCA1 in neurons (Aging (2017)). The showed that overexpressed BRCA1 mitigates PDS-induced DNA damage. Similarly, in the current work does overexpression of ATG7 enzyme modulate the PDS-related phenotypes observed?

The experiments to address this question were technically challenging because our bioinformatics analyses revealed the presence of putative G4-DNA motifs in virtually all autophagy genes. However, importantly, many autophagy genes contain just a few putative G4-DNA motifs, whereas the *Rattus norvegicus Atg7* gene contains 27. As a result, transcription of *Atg7*, which diminishes with aging (Lipinski et al., 2010; Lu et al., 2004, can potentially be more sensitive to the G4-DNA ligands than transcription of other genes (autophagic and non-autophagic genes). These questions were not addressed in the current study and are currently being pursued in our lab. Nevertheless, for the resubmission, we cloned the cDNA of *Atg7*, which lacks a number of putative G4-DNA motifs located in the introns, including the 2700 G4-DNA under investigation. We applied single-cell longitudinal analysis to gain spatiotemporal resolution and to simultaneously visually monitor the accumulation of p62, an autophagy substrate and autophagic marker, and neuronal toxicity in neurons expressing ATG7-mApple treated with a vehicle or PDS. Indeed, we observed a rescue effect of overexpressed ATG7-mApple on the degradation of p62-GFP in the presence of PDS. We added the data to the revised manuscript (Figure 5—figure supplement 3).

Is DNA damage accumulation in neurons also observed in the current work? How are the results and findings from the two studies related, if at all?

We thank the referee for this comment. We think this is a very important one. In our manuscript, we did not aim to conclude that the only way G4-DNA causes neurodegeneration is by downregulating autophagy. In our previous work, we showed that stabilizing G4-DNA leads to lower levels of *Brca1* andBRCA1 and, as a result, to accumulation of DNA damage (Moruno-Manchon et al., 2017). In the current manuscript, we showed that stabilizing G4-DNA strongly downregulates *Atg7*, ATG7, leading to reduced autophagy and neurotoxicity. At the same time, we cannot exclude the possibility that more factors may contribute to neurodegeneration. With that in mind, we agree with the reviewer and have discussed this issue in the manuscript.

Autophagic genes can be epigenetically silenced (Artal-Martinez de Narvajaset al., 2013; Baek and Kim, 2017; Lapierre et al., 2015). However, epigenetic silencing is a common mechanism in aged cells and affects the genes well beyond the autophagy pathway genes. In many models, the DNA damage repair genes and other gene types are epigenetically silenced in cancer and during aging (Lahtz and Pfeifer, 2011; Langie et al., 2017; Liu, Yip and Zhou, 2012). Our work suggests that an age-associated change in DNA conformation could be a *novel epigenetic-like mechanism* of gene expression in aging neurons; therefore, the results from our two studies are related. Please see the fourth paragraph in the Discussion section that describes this issue.

Reviewer #2:In this paper, the authors rather convincingly show that the ATG7 gene, which is critical for the initiation of autophagy and whose transcription decreases with aging, does contain a bona fide G-quadruplex (G4) in its first intron and that stabilization of G4 using pyridostatin (PDS), a well-known benchmark G4 ligand, downregulates this ATG7 gene. All this suggest that stabilization of ATG7 G4 does interfere with the transcription of this gene, thereby inhibiting induction of autophagy. In good agreement, the authors found that mice treated with PDS develop memory deficits and accumulation of lipofuscin, suggesting premature aging and deficient autophagy. Moreover, brain samples from aged mice contain G4-DNA which are absent in brain samples from young mice. Finally, the authors showed that overexpressing the helicase Pif1, which is known to resolve G4, in neurons exposed to PDS improves the various phenotypes associated with PDS treatment, thereby suggesting that G4-DNA may represent an interesting and relevant intervention point for boosting autophagy and thereby interfering with neurodegeneration.In general I think this is a very interesting study based on well-designed and well-conducted experiments that deserves to be published in eLife. In particular, revealing that G4 may represent a new putative intervention point to interfere with autophagy-related neurodegeneration represents an important discovery. Also, the idea that G4-ligands may induce/accelerate neurodegeneration is highly important, especially with respect to their putative development to the clinics in other field like cancerology. However, I have a few comments that need to be addressed before this paper could be deemed for publication in eLife.

We thank the reviewer for these comments.

- Figure 1: the search for PQFS in the gene and promoter sequences of autophagy genes. The analysis performed by the authors indicates that all the autophagy genes contain putative G4-DNA. It would be important to precise the percentage of total genes that contain, or not putative G4-DNA to assess how specific is this correlation.

We fear that we might not understand the referee’s point. Here, we analyzed genes commonly described as “autophagy-related genes” in the autophagy literature. That means that other genes, even those that can potentially regulate autophagy directly or indirectly, were not taken into account in our analyses. We would like to point out that all of those genes were already analyzed for putative G4-DNA motifs by other labs Chambers, 2015).

- Figure 2: 2 µM PDS has a much stronger negative effect than 2 µM BRACO19 on the level of the ATG7 transcript. However, the effect on ATG7 protein level is similar for both compounds suggesting that BRACO19 could also interfere with translation of ATG7 mRNA or on ATG7 protein stability. The authors should comment on this.

We thank the reviewer for this comment. The interaction mechanisms between the G4 structures and PDS or BRACO19 differ in that BRACO19 is considered a “pan-quadruplex” ligand as it binds to quadruplexes whatever their secondary structures, and PDS is more selective for parallel-type quadruplexes (Ruggiero and Richter, 2018). Therefore, the fact that BRACO-19 and PDS affect levels of the *Atg7* transcript differently but levels of the ATG7 protein similarly could originate from selectivity of drugs for G4 structures. This point has been added to the discussion (please see the fifth paragraph in the Discussion section).

As for BRACO19, although it is very important to use another GA-ligand than PDS to support the authors' main conclusions, it is somehow surprising that BRACO19 is used only in this experiment and also that its putative stabilizing effect on ATG7 G4-DNA is not properly assessed (see below).

The reviewer is correct, of course: we should have used the BRACO19 ligand in other experiments as well (please see Figure 2, Figure 3 and Figure 4—figure supplement 1, Figure 5—figure supplement 1). BRACO19 downregulated ATG7 and Atg7, “slowed” melting Atg7-32, promoted the formation of N-TASQ puncta, inhibited autophagy and mitophagy.

- Figure 3: The CD signature of the ATG7 gene in the classical conditions is not fully convincing and could correspond to a mix of several structures (not only G4). The authors argue that the presence of dehydrating agents (PEG, CH_3_CN) reduces the polymorphism and indeed the CD looks better but this effect has been reported only for telomeric sequences and is not fully admitted. Therefore, the salt effect (Li+ to K^+^) should preferably been tested using the HF2 antibodies.

We agree that the use of dehydrating conditions to improve the CD signature of a quadruplex structure has not been examined thoroughly despite an initial impetus provided by Chaires et al.,2013). However, we also reported this effect for non-telomeric quadruplexes, extending this observation to quadruplex-forming sequences in the promoter of human genes (e.g., MYC, cf. Monchaud et al., 2014). To further address the reviewer’s concerns about the ability of the ATG7 sequence to fold into a quadruplex structure, we performed a new series of experiments and added the data to the revised manuscript (Figure 3). Briefly, we performed an NMR experiment with ATG7-32 sequence ((d[GGGGCTGGGGTCCCTTGGGGGAACTGTATTGGG]) and compared the results with ATG7-32mut (d[GCGCCTGCGCTCCCTTGCGCAACTGTATTGCG]), a sequence that cannot fold into quadruplex. For the HF2 binding experiment, we determined the effects of K^+^ (known G4 stabilizing cation) and Li^+^ (known G4-destabilizing cation) and saw clear, expected results (Figure 4A,B).

**Author response image 2. respfig2:** 

The results seen in Author response image 2 clearly indicated that ATG7-32 folds into a complex structure comprising both a quadruplex core (^1^H-NMR signals between 10 and 12 ppm) and duplex stems (^1^H-NMR signals between 12.5 and 14 ppm), which explains the complicated CD signature, whereas ATG7-32mut displays only duplex stem contribution. Of note, the polymorphism of the ATG7-32 quadruplex is also obvious, given the low resolution of the signal in the region of the NMR signals typical of the quadruplex core. Again, these results further substantiate the complicated CD signature observed in ‘native’ (that is, without dehydrating agent) conditions.

In line with what I discussed just above, BRACO19 should also be tested in the UV-melting experiments performed in panel c to determine if it also stabilizes, or not, the most probable ATG7 G4 and also in the experiments performed with N-TASQ on neurons in panels I and J.

We thank the reviewer for this comment. We added the data to the revised manuscript (Figure 3, Figure 4—figure supplement 1).

The TDS spectrum without dehydrating agents should also be shown. TDS and CD spectra should be shown on different panels and the CD in K^+^ conditions should not be termed "cont" (for control?) as it is misleading. Also, another classical control for CD which is missing is the use of scrambled G runs.

We agree and apologize for this oversight. We performed the requested experiments and presented the data splitting the CD and TDS results (left and right panel, respectively; Figure 3C,D).

The results seen in Figure 3 indicate clearly the topological differences between ATG7-32 and its mutated counterpart, the latter being characterized by both a CD signature (cf. Vorlickova et al., 2009) and a TDS signature (cf. Mergny et al., 2005) typical of a duplex structure with high GC content.

Another important point is that the in vitro experiments presented in panels e to h should be repeated in presence, or not, of PDS or of BRACO19 to determine if these two G4 ligands may, or not, interfere with the binding of the HF2 antibody and/or of G4-binding proteins. This point looks important to me as the general assumption is that G4-ligands stabilize G4 structures but, in principle they could also destabilize them, and/or even prevent the binding of various factors such as antibodies or G4-binding proteins by direct competition.

We thank the referee for this comment. We fear, however, that it is not easy to address their concern. First, the precise mode of binding of the antibody HF2 to G4s is not known. HF2 might discriminate between various quadruplexes from the KIT gene (cf. Balasubramanian et al., 2008), but the lack of solid structural data precludes any attempt to rationalize the way it binds to quadruplex. Second, it has also been demonstrated with another quadruplex-selective antibody, BG4, that the concomitant binding of both an antibody (BG4) and a ligand (cPDS) is possible (cf. Balasubramanian et al., 2015). Third, examples of quadruplex-destabilizing agents are still very sparse in the literature, and reported examples (e.g., TMPyP4, cf. Pearson et al., 2014) have been collected under debatable experimental conditions (e.g., very high ligand/DNA ratio), far from any biological relevance. Therefore, performing experiments in presence of both antibodies and ligands is possible, but the many possible outcomes (e.g., competition, cooperation) make the exploitation of the results far too complex to be attempted. Nevertheless, we again thank the reviewer for this comment and will address this in future experiments.

Finally, as for the experiments with N-TASQ, I have some problem to understand them because, as this fluorescent molecule is also a G4-ligand, one may imagine a competition between this compound and PDS for the binding on G4. And, indeed, in the original paper on N-TASQ (Laguerre et al., 2016), this problem is discussed and addressed (by using BRACO19 instead of PDS) and the authors concluded that at high concentration (100 µM) N-TASQ provides high resolution images but does not allow to visualize significant differences between BRACO19-treated and -untreated cells and that the only conditions that allowed to visualize an increase in N-TASQ stained nuclear foci was a low dose of N-TASQ (2.5 µM) and of BRACO19, and that a higher dose of BRACO19 leads to a BRACO19 dose-dependent decrease in N-TASQ staining, presumably because of a competition between these two G4-ligands for binding on G4. Hence it is hard for me to understand why and how PDS treatment should lead to an increase in N-TASQ staining (used at 50 µM here) as shown in panel i and j. Of note such a competition with PDS has been described for DAOTA-M2, another G4-specific fluorescent probe (Shivalingam et al., 2015). All this should be discussed and the effect of BRACO19 on N-TASQ staining should also be tested.

We thank the reviewer for this comment and apologize if our explanation was not clear in the initial manuscript. Briefly, the most important parameters to control are the live-cell incubation of the quadruplex ligand (e.g., PDS, BRACO19 – with ad hoc concentrations) and the post-fixation cell labelling with N-TASQ (at a concentration that must be fine-tuned). This protocol, which allows for assessing the extent (and modification) of the quadruplex landscape upon ligand treatment, was developed with MCF7 cells (cf. Monchaud et al., 2016) and subsequently validated in HeLa cells (in presence of either BRACO19 or TMPyP4, cf. Monchaud et al., 2017). Importantly, this protocol was used with cancer cells but never with neurons, and the collected results presented here lend further credence to its reliability and broad applicability.

- Figure 4: BRACO19 should also be tested in at least one of the experiments presented here as there are at the basis of the main conclusion of the paper, destabilization of DNA-G4 represents a relevant and interesting intervention point ATG7 to interfere with neurodegeneration associated with aging-related decline in induction of autophagy.

As discussed above, there are no reliable examples of small molecules reported as quadruplex-destabilizing agents, apart from somewhat debatable candidates. Specifically, the role of BRACO19 in destabilizing G4-DNA remains questionable in light of the many recent reports (e.g., over 50 articles published on BRACO19).

- In the Discussion section, my suggestion is that the authors should discuss about the ability of G4-ligand to stabilize DNA-G4. Is it a general property of all the G4-ligands or is it specific to a subset of G4-ligands (that includes PDS and BRACO19)? Should a compound that efficiently bind G4 without any effect on their stability exist, then it would represent an ideal control to further validate their findings. Also, the possibility that the PDS-related phenotypes may involve its effect on RNA-G4 should be mentioned and discussed.

We thank the reviewer for their comments. We substantially expanded the Discussion section. Please see the fifth paragraph of the Discussion section.

To finish, and importantly, in my view the main message of this manuscript is that G4-DNA may represent an interesting and relevant intervention point for boosting autophagy and thereby interfering with neurodegeneration, rather than the discovery of a novel pathway that regulates autophagy in neurons, as stated by the authors already in the title. Indeed, if the authors want to state that their findings do reveal a novel pathway for regulating autophagy, than they need to find physiological situations where the stability of DNA-G4 present in autophagy genes (in particular in ATG7) may be tuned by various cellular pathway(s)/component(s) which, this way, regulate autophagy. As for now, they essentially showed that stabilizing G4 using PDS downregulates ATG7 and that PDS induces memory deficits and autophagy in mice and that, on the contrary, overexpressing the G4-DNA helicase Pif1 in neurons exposed to PDS suppresses PDS-associated phentotype. Not to mention that this G4 stabilization/PDS effect may also be at the level of G4-RNA. Therefore, I suggest that the authors down tune their message, especially in the Title but also in the discussion. I guess that revealing a new and relevant intervention point for modulating autophagy in neurons is per se sufficiently interesting in addition to be of biomedical relevance.

We thank the referee for these comments. In our original submission, we showed that (1) stabilizing G4s downregulates *Atg7*, ATG7, and autophagy; (2) old brains contain more G4s but young brains hardly any; (3) *Atg7* is downregulated in old brains; (4) Pif1 rescues phenotypes associated with PDS treatment (e.g., autophagic phenotypes); and (5) the expression of Pif1 itself slightly upregulates autophagy in neurons. We, therefore, speculated in the Discussion section that in addition to histone acetyltransferases facilitating chromatin decondensation and promoting the expression of autophagy-related genes (Baek and Kim, 2017; Lapierre et al., 2015), Pif1 may help to sterically allow the transcriptional machinery to transcribe DNA. To respond to the reviewer’s conceptual concerns, we performed additional experiments with an ATPase/helicase-dead mutant version of Pif1 (also requested by reviewer 1) and added the results to the revised manuscript (Figure 8F). Mutated Pif1 was not able to rescue autophagic phenotypes in neurons. Therefore, while we are delighted that the reviewer is accepting of the main conclusions of our study, we respectfully disagree with the reviewer’s assessment of “the main message.” We believe that our study revealed a new epigenetic-like mechanism of autophagy regulation in neurons. Our conclusions reflect the prevailing views in the autophagy field (Lapierre et al., 2015), as the field is highly interested in understanding *why autophagy is diminished in neurons with aging*, leading to age-associated neurodegenerative disorders. Nevertheless, we have taken this opportunity to review our description/discussion of these findings and to revise them to more effectively explain the main conclusions without overstating the significance.

Reviewer #3:The authors used the G4 stabilizer pyridostatin (PDS) in cultured neurons and in mice to unravel a role for G4-DNA structures during Atg7-mediated autophagy. They used biophysical methods to demonstrate accumulation of G4-DNA in the Atg7 gene and immunostaining methods to illustrate an age-dependent global increase of G4-DNA. The authors further show that the majority of phenotypes induced by PDS were partially rescued by overexpression of fhe Pif1 helicase.The manuscript is clearly written, and data are well presented. The implication of G4-DNA in autophagy and age-related neurological deficit is novel and will be of interest to a broad readership.

We thank the referee for these comments.

It is not, however, clear if the effects reported here are direct consequences of G4-DNA in the Atg7 gene or indirect global perturbations of G4-DNA homeostasis. PDS is a blunt tool potentially impacting over 600,000 putative G4-DNA structures. Also, Pif1, as acknowledged by the authors, impacts telomere length, so how can the authors be sure that the reported partial improvements of neuronal phenotypes are solely due to resolving G4-DNA structures? Is the partial rescue specific to G4s in the Atfg7 gene?

Reviewer 1 raised a similar question. Initially, we viewed G4s as pathogenic DNA and RNA structures that occur only in some neurodegenerative diseases (e.g., frontotemporal dementia and ALS (Haeusler, Donnelly and Rothstein, 2016). However, analyses of the aged brain samples with *no* neurodegenerative pathologies led to a surprising and serendipitous finding—that the G4 structures are present in the aged brains and could be a marker of senescence in aging cells (e.g., as DNA methylation or aneuploidy or transposable element dysregulation in neurons (Mosch et al., 2007; Fischer et al., 2012; Sun et al., 2018)). By the end of the study, we understood that the Pif1 helicase can partially rescue “aging” autophagic phenotypes. Therefore, at a larger scale, we view our data on an age-associated change in DNA conformation as a novel epigenetic-like mechanism of gene expression in aging neurons. Recently, transposable elements were found to be epigenetic regulators of the genome, opening a new avenue of exciting research in neurodegeneration (Sun et al., 2018). We believe that G4-DNA is an additional layer for such epigenetic-like regulation. As an autophagy lab interested in neuronal autophagy (Moruno Manchon et al., 2016; Tsvetkov et al.,2013; Moruno Manchon et al., 2015; Moruno Manchon et al., 2016), we started investigating the roles of G4-DNA in neuronal aging with assessing their role in autophagy. Please see the fourth paragraph in Discussion section, which discusses this important issue.

Neurons are post-mitotic and how telomere length could be affected by Pif1 is not exactly clear. Indeed, cell-cycle activity is a driving force for telomere shortening. In some neurodegenerative diseases and in advanced aging, post-mitotic neurons re-enter the cell cycle, leading to various DNA abnormalities (Mosch et al., 2007; Fischer et al., 2012). Neurons with a shorter telomere length can be generated from aged fibroblasts (Huh et al., 2016. We believe it is unlikely that Pif1 is neuroprotective due to a telomere effect. As Pif1 is a G4-DNA helicase and its function to unwind G4-DNA, its beneficial effects are likely due to a helicase activity. Indeed, in the revised manuscript, we show that a helicase dead mutant lost its effect (Figure 8F). Nevertheless, we cannot fully exclude the possibility that Pif1 may have additional unknown functions besides being a G4-DNA helicase.

Above we explained our motivation for studying the *Atg7* gene (please see our response to reviewer 1’s comments). Briefly, among ATG-related genes, *Atg7* contains one of the highest amount of putative G4 motifs, and it works at the very beginning of the initiation of autophagy. As we believe that G4-DNA functions by means of an epigenetic like mechanism, Pif1’s rescue effect is not specific to *Atg7*.

Performing a classical epistasis experiments is critical in this manuscript. For example, repeating key experiments in presence and absence of Atg7 will confirm that the reported phenotypes are due to direct modulation of G4-DNA in the Atg7 gene, hence supports the conclusion of a novel pathway as stated in the Title.

We thought this was an excellent suggestion. We collected new data, which are now added to the revised manuscript, to address the reviewer’s suggestion (Figure 5—figure supplement 3).

For the huntingtin experiments in Figure 4, I suggest using patient derived fibroblasts or iPS-derived striatal neurons instead of ectopic expression of the exon-1 fragment of the poly-Q huntingtin. Although, in silico predictions using the QGRS mapper rule out G4-DNA, it is important to experimentally rule it out in the native genomic environment.

As we study autophagy in the lab, we frequently use mutant huntingtin as a marker of autophagic clearance effectiveness. Huntingtin is a big protein (>3,300 amino acids), and its gene contains numerous putative G4-DNA motifs. Its half-life is quite long, however, and so, we can assess huntingtin degradation before PDS-associated effects on the huntingtin gene’s transcription. We cultured primary neurons from BACHD mice to determine if PDS affects the levels of mutant huntingtin. BACHD mice from the Yang lab (UCLA) express a human mHtt (97Qs) genomic locus and flanking sequences. The BACHD model recapitulates many of the molecular/cellular/behavior features seen in Huntington disease patients (Gray et al., 2008). PDS promoted accumulation of mutant huntingtin indicating that the degradative pathways in neurons are affected by PDS (Figure 5—figure supplement 2).

The protein p62 is a known hallmark of perturbed autophagy. Is p62 aggregation also modulated by PDS in an Atg7 dependent manner? This is important since this hallmark protein aggregation is a common phenomenon in a number of age-associated neurological disorders including Huntington's and ALS/FTD. In the opinion of this reviewer, this is a better hallmark of perturbed autophagy given its clinical relevance.Overall, the concept is novel and exciting but the data in its present form do not support the main conclusion.

We thought this was a great suggestion. We collected new data, which are now added to the revised manuscript (Figure 5—figure supplement 3). We are pleased that this reviewer accepts the novelty of our study and hope that the new data support the conclusions of our manuscript.

Additional References:

Dempsey, W.P., Fraser, S.E. & Pantazis, P. PhOTO zebrafish: a transgenic resource for in vivo lineage tracing during development and regeneration. PLoS One 7, e32888 (2012).

[Editors’ note: what follows is the authors’ response to the second round of review.]

The reviewers have discussed the reviews with one another and the Reviewing Editor has drafted this decision to help you prepare a revised submission.Summary:This manuscript is a resubmission of a previous one in which authors show that the G-quadruplex (G4) ligand pyridostatin (PDS) was found to downregulate expression of the Atg7 gene in neurons. The first intron of the Atg7 gene contains predicted G4-forming sequences that seem to form G4 and interact with PDS. Mice treated with PDS develop memory deficits and accumulation of lipids and proteins previously observed to accumulate in aged brains. Brain samples from aged mice, but not young mice, contained G4 DNA, and overexpression of the G4-resolving helicase Pif1 in neurons improved the phenotypes associated with PDS treatment. Based on their findings, the authors conclude that G4 DNA is involved in regulating autophagy in neurons. The authors have satisfactorily responded to the concerns raised by the referees, but q few points need to be taken before the manuscript can be accepted.Essential revisions:- The 3 quartet structure shown in Figure 3 has a low probability of formation due to the presence of 3 long loops (5-nt, 7-nt, 9-nt) which drastically reduce its stability (see various methods of the G4 score calculation in Bedrat et al., 2016; Puig-Lombardi et al., 2019). Hence it follows that the Atg7-32 sequence is most probably highly dynamic and may form several secondary structures that exist in equilibrium (various G4, hairpins etc.), which is good agreement with the very broad profile of the NMR spectra. Hence this analysis does not allow the authors to firmly conclude the existence of a stable G4. Therefore, the authors should down-tune, or at least modulate their G4 hypothesis.

The referees correctly point out that we only briefly described how Atg7-32 may fold into several secondary structures that likely exist in an equilibrium. Therefore, we reviewed our explanation of the results and revised the manuscript to more effectively elaborate on how Atg7-32 may form various structures. In particular, we re-wrote the paragraph that explains the observed data and it now reads as the following.

“We further investigated the higher-order structure of both Atg7-32 and mutAtg7-32 by nuclear magnetic resonance (NMR). Both displayed ^1^H-NMR signals in the 12–14 ppm region, which corresponds to duplex stems (providing a rationale for the complicated CD/TDS signature of the former), but only Atg7-32 had ^1^H-NMR signals in the 10–12 ppm region, characteristic of a G4-DNA structure (poorly defined here, indicating a mixture of G4 topologies) (Figure 3E). These signals indicate that Atg7-32 may fold into a variety of G4-DNA topologies, including both 3- and 4-G-quartet G4s with both short (2-nt) and long (9-nt) hairpin-forming loops (Figure 3B), which were also detected earlier in non-neuronal cells^3,40^ or computationally predicted^41,42^. An equilibrium among all these various topologies is illustrated by the complex signatures generated with CD, TDS and NMR.”

In addition, we removed the word “stable” from the following sentence in the Discussion: “We showed that a PQFS identified in the Atg7 gene can fold into a G4 structure, as demonstrated by spectroscopy (CD, TDS and NMR) and its interaction with PDS and BRACO-19, the HF2 antibody, and the G4-binding protein PC4”.

We also thank the referees for the references. In addition to citing Bedrat et al., 2016 and Puig-Lombardi et al., 2019, we now included a citation for Chambers et al., 2015, as Chambers et al., specifically emphasized the following: “To understand the potential functions of G4s, we quantified the prevalence of OQs in genomic regions associated with promoters, 3′ and 5′ untranslated regions (UTRs), exons, introns and splicing junctions (Supplementary file 4). Notably, a large proportion of these regions (up to 49% in PDS and 46% in K^+^) compose exclusively noncanonical G4s (i.e., long loops or bulges)”. Please see the new Figure 3B that illustrates these possibilities.

- The fact that Pif1 rescues PDS-induced phenotypes in cultured primary neurons (Figure 8). This observation is interesting but somehow a bit surprising and rather counter-intuitive as it is not fully consistent with numerous studies reported in the literature that show that the G4 unwinding activity of most of the G4 helicases is indeed prevented by G4 ligands. This has been shown in particular for Pif1 (see Mendoza et al., 2016; Mergny et al., 2015; Balasubramanian et al., 2015 plus references cited therein). Therefore, the assumption that Pif1 rescues PDS-induced phenotype by unwinding G4 in the Atg7 is unclear. The authors should discuss the results of their experiment with Pif1 in light of all the published data indicating that Pif1 G4 unwinding activity is inhibited by various G4 ligands that include PDS, or alternatively they could test their hypothesis by performing a functional in vitro assay (e.g.: unwinding assay with or without PDS).

We thank the referees for this comment. That is an excellent suggestion, as we indeed did not discuss that several prior studies investigated how an unwinding activity of G4 helicases is affected by G4-ligands (e.g., PDS). We did not emphasize that several previous studies reported that, in general, G4-ligands impede helicase functions. In our study, we show the opposite: the PIF1 helicase rescues PDS-associated phenotypes in living neurons.

We would like to point out that others published exciting studies on the helicase/G4-ligand assays focused on in vitro systems only. Therefore, the relevance of their findings to our study is not that straightforward. For example, these in vitro studies used an excess of the G4-ligands that may have a strong effect on the outcome.

Critically, these studies used a G4-forming sequence without its complementary sequence in the in vitro assays. In these assays, adding the complementary sequence (so called the “trap” oligonucleotide) fully unfolds the G4-DNA/ligand complexes (Mendoza et al., 2016). In contrast, our study focused on how PIF1 rescues PDS-associated phenotypes in living neurons. Our in vivo context and low concentrations of PDS may explain why our data are different from those generated in vitro. Finally, in our neuronal system, primary neurons are transfected with PIF1, and PDS is later added—after PIF1 becomes overexpressed. We have taken this opportunity to review our discussion of the literature and to revise the manuscript to meticulously explain the main conclusions, as follows.

“Intriguingly, prior in vitro studies found that Pif1’s G4-DNA unwinding activity is diminished by G4 ligands (e.g., PDS), which appears to contradict to our in vivo findings. Nevertheless, the relevance of these data to our study is not straightforward since a G4-DNA forming sequence was used without its complementary sequence in the in vitro studies. Adding the complementary DNA sequence unfolds the G4-DNA/ligand complexes^84^. In addition, the in vitro experiments assayed the activity of Pif1 using an excess of G4 ligands^84^, and therefore, the data are not easy to extrapolate to our neuronal in vivo model. Also, in our studies with living neurons, Pif1 was overexpressed before PDS was added to the media, and thus, the kinetics of Pif1-G4-DNA-PDS interactions may be overly complex for a direct comparison to the in vitro conditions.”